# Development of an evidence-based complex intervention for community rehabilitation of patients with hip fracture using realist review, survey and focus groups

Jessica Louise Roberts,[1] Nafees Ud Din,[1] Michelle Williams,[1] Claire A Hawkes,[2] Joanna M Charles,[1] Zoe Hoare,[1] Val Morrison,[3] Swapna Alexander,[4] Andrew Lemmey,[5] Catherine Sackley,[6] Phillipa Logan,[7] Clare Wilkinson,[1] Jo Rycroft-Malone,[1] Nefyn H Williams[1,4]

For numbered affiliations see end of article.

**Correspondence to**
Dr Jessica Louise Roberts;
j.l.roberts@bangor.ac.uk

## ABSTRACT

**Objectives** To develop an evidence and theory-based complex intervention for improving outcomes in elderly patients following hip fracture.

**Design** Complex-intervention development (Medical Research Council (MRC) framework phase I) using realist literature review, surveys and focus groups of patients and rehabilitation teams.

**Setting** North Wales.

**Participants** Surveys of therapy managers (n=13), community and hospital-based physiotherapists (n=129) and occupational therapists (n=68) throughout the UK. Focus groups with patients (n=13), their carers (n=4) and members of the multidisciplinary rehabilitation teams in North Wales (n=13).

**Results** The realist review provided understanding of how rehabilitation interventions work in the real-world context and three programme theories were developed: improving patient engagement by tailoring the intervention to individual needs; reducing fear of falling and improving self-efficacy to exercise and perform activities of daily living; and coordination of rehabilitation delivery. The survey provided context about usual rehabilitation practice; focus groups provided data on the experience, acceptability and feasibility of rehabilitation interventions. An intervention to enhance usual rehabilitation was developed to target these theory areas comprising: a physical component consisting of six additional therapy sessions; and a psychological component consisting of a workbook to enhance self-efficacy and a patient-held goal-setting diary for self-monitoring.

**Conclusions** A realist approach may have advantages in the development of evidence-based interventions and can be used in conjunction with other established methods to contribute to the development of potentially more effective interventions. A rehabilitation intervention was developed which can be tested in a future randomised controlled trial (MRC framework phases II and III).

### Strengths and limitations of this study

► A complex intervention for hip fracture rehabilitation was developed, which was evidence based and theoretically underpinned (Medical Research Council (MRC) framework phase I).

► Programme theories were developed from a realist review of the literature. A survey added context, and focus groups provided data on the experience, acceptability and feasibility of rehabilitation interventions.

► The methods used to develop this rehabilitation programme may be applicable to the development of other complex interventions.

► The feasibility and acceptability of the developed intervention are reported separately, but evidence of effectiveness and cost-effectiveness requires testing in a randomised controlled trial (MRC framework phases II and III).

**Trial registration number** ISRCTN22464643, Pre-results.

## BACKGROUND

Proximal femoral fracture, more commonly referred to as hip fracture, is a common, major health problem in the elderly.[1] It is associated with prior fragility fracture, and other comorbidities such as: cognitive impairment, undernutrition, decreased bone mineral density, frailty, poor physical functioning, vision problems and weight loss.[2] Mortality is high with 14%–58% of patients dying within 12 months[3 4] and up to 53% do not regain their previous level of functioning.[5 6] Proximal femoral fractures cost the UK economy approximately £2.3 billion a year.[7] Management guidelines from the National Institute

of Health and Care Excellence (NICE)[8] recommend multidisciplinary rehabilitation, which has the potential to maximise recovery, enhance quality of life and maintain independence. However, systematic reviews conclude that there is insufficient evidence of overall effectiveness or cost-effectiveness, but individual components of such programmes may show promise.[9–11]

Rehabilitation programmes for hip fracture are complex interventions due to their multifaceted nature and the involvement of many heterogeneous factors including individual patient circumstances and comorbidities, healthcare professionals, rehabilitation setting and social influences.[12] The interaction of these factors in the real world and how they interplay and influence each other to determine the success and failure of such programmes is poorly understood, making it difficult to identify which specific components of rehabilitation programmes are effective and under what circumstances.[13 14] While there have been many systematic reviews of hip fracture rehabilitations,[5 10 11] these are only able to evaluate the evidence of whether an intervention works and do not allow for exploration of how and why an intervention leads to its reported outcomes. Realist reviews aim to elucidate the mechanisms behind an intervention and determine 'what works, for whom, in what circumstances, and why?' while taking into account the heterogeneous nature of such interventions and the settings in which they are delivered.[15] This involves multiple steps, which starts with extracting working theories from individual studies and developing them into 'programme theories' which describe what programmes or interventions are expected to do and how they are intended to work. These are compared and contrasted to develop intermediate programme theories, which refer to propositions of how a programme is likely to produce intended outcomes. These are then tested and refined into a final list of theories, which describe the mechanism or causal force that makes things happen in certain circumstances or contexts, such as patient characteristics or place of rehabilitation, which result in desired outcomes.[13]

Realist reviews provide a flexible way of exploring causal relationships, thus aiding our understanding of intervention mechanisms and supporting the development of potentially more effective interventions.[16 17]

We therefore undertook a realist review of the available evidence for hip fracture rehabilitation to develop theory on the context, mechanism and outcomes of existing rehabilitation programmes, with this forming the basis for the development of our own evidence-based intervention for subsequent testing in a feasibility study.[18] The development of these theory areas was performed in conjunction with a survey of current practice by UK rehabilitation health professionals and focus groups with patients, carers and multidisciplinary rehabilitation team. While the Medical Research Council (MRC) framework for complex interventions provides general guidelines for intervention development and supports the use of theoretical underpinning,[12] detailed guidance on how this

framework is practically applied to intervention development is lacking. To contribute to bridging this knowledge gap, this paper sets out the methodology of how the evidence base was established and used for intervention development, linking the findings of the review, survey and focus groups to the proposed aims of our intervention and how we expected these to facilitate our intended outcomes.

## METHODS

A summary of the methods used is presented below. Further detail can be found in the final report to the funder. The development of the community-based rehabilitation package was informed by three complementary work packages. A coherent theoretical basis for the intervention was developed from a realist literature review. Initial findings from the review were used to develop the questions for a survey. A survey of current services described usual practice, and was an additional source of relevant theories that contributed to the realist review. The initial findings from the review were also used to develop topic guides for the focus groups and the initial framework for their analysis. Focus groups provided data on the experience, acceptability and feasibility of rehabilitation interventions.

## REALIST REVIEW

A realist review was used to identify the evidence base and develop a theoretical understanding. A scoping search of systematic reviews,[5 10 11 19–53] guidelines[8 54–58] and theoretically rich primary studies[59–64] was performed to map out the important areas and research gaps (NUD, NHW). This generated a list of questions, which could be grouped under different domains relating to: patients, healthcare and rehabilitation teams, rehabilitation programmes and the settings in which rehabilitation was delivered.

The list of questions was formulated into statements that described how these different domains interact. These statements were subsequently refined during discussions between members of the research team (JRM, NUD, NHW, JMC) and with other researchers from the School of Healthcare Sciences, Bangor University, in two realist review workshops.

Feedback from experts in health psychology, rehabilitation and implementation research were combined with initial survey and focus group findings into candidate programme theories. These intermediate working theories were used as the basis for bespoke data extraction forms.

## LITERATURE SEARCH

The literature search strategy used by the NICE guideline review of multidisciplinary rehabilitation programmes for hip fracture[8] was adapted for this review, with no search filters for study design; this intentional inclusivity enabled

| Table 1 | Working definition of multidisciplinary rehabilitation used to screen sources of evidence |
|---|---|
| Purpose | Supports re-enablement of the frail elderly (over 65 years old) following proximal hip fracture to achieve their functional potential and maintain independent living where possible |
| Functions | A bridge between: (A) the hospital and the community; (B) different healthcare sectors and personal social care<br>Views people holistically<br>Time limited (up to 1 year following fracture) |
| Structure | Teams based in hospitals, the community or across both sectors |
| Content | Treatment and therapy (to increase strength, confidence and ability to perform activities of daily living)<br>Psychological, practical and social support<br>Support/training to develop skills and strategies for self-management |
| Delivery | Care delivered by a multidisciplinary team or teams |

review of different types of study, which, in turn, facilitated formulation and examination of the emerging theories. Twenty-one databases were searched from inception to February 2013 in order to be used as the next step of programme development (online supplementary appendix 1). Citation tracking and internet search engines were used to identify additional evidence as the review progressed and as new ideas emerged. Materials were retrieved purposively to answer specific questions or test specific theories until no new themes emerged.

### Screening and categorisation of references
Participants of interest were elderly adults with proximal hip fracture. The intervention of interest was multidisciplinary rehabilitation following proximal hip fracture. Outcomes of interest were mortality, pain, functional status, quality of life, health utility, health service use, costs and patients' experiences. A working definition of multidisciplinary rehabilitation was adapted from a review of intermediate care services[65] (table 1). Separate reviewers screened identified studies for relevance and discrepancies were resolved after discussion (NUD, NHW, JMC). Potentially relevant papers were categorised according to study type, and then according to whether they were conceptually 'rich', 'thick' or 'thin'[66 67] (NUD, NHW). We started by extracting data from all of the 'rich' studies and sampled data from the conceptually 'thick' studies until saturation was reached. The purpose of this was to make the database manageable and to build and examine theories from studies with the most relevant concepts.

### Data extraction and quality assessment
Data were extracted by one reviewer (NUD) and checked for accuracy by a second (NHW). We assessed study quality using the mixed methods appraisal tool.[68] Data from effectiveness studies were exported into structured tables to show the strength and direction of the treatment effects.

### Testing the theories with quantitative and qualitative evidence
Theories were refined through an iterative process comparing individual study programme theories in turn. Data for each individual study were examined in terms of the identified programme theories and the interaction between mechanisms, context and outcomes, starting with data extracted from studies that were conceptually 'rich' and continuing with those that were conceptually 'thick'. The data were then examined across the different studies to detect patterns and themes. A second set of refined data extraction forms was used to test each theory in turn, and adjudicate between components of the final programme theories.

## SURVEY OF UK HIP FRACTURE CENTRES
### Survey development
A UK-wide web-based survey was conducted, targeting physiotherapists, occupational therapists and hip fracture centre therapy service managers working in the rehabilitation of patients over 65 years of age who have had surgery for proximal hip fracture. NICE guidance on hip fracture rehabilitation[5] was used with initial findings from the scoping review as the starting point for developing the questions. The questionnaires were piloted on members of staff across one health board in Wales, and minor amendments were made.

### Data collection
Three versions of the survey were developed for hip fracture centre managers, physiotherapists and occupational therapists. The managers' survey focused on the organisation of services, while the therapists' questionnaires focused on aspects of clinical practice such as session content, frequency and location, and how assessments were conducted. The therapists' versions were further subdivided according to healthcare setting: acute hospital, community hospital or community-based team.

The survey was open for 7 weeks from 6 August 2013 to 25 September 2013. We surveyed a sample of senior managers who had a strategic role in rehabilitation services for this group of patients and aimed to achieve a 10% sample of all UK centres performing hip fracture surgery. Centres in Wales, Northern Ireland and England were identified from publicly available information on the National Hip Fracture Database (NHFD). Hospitals in Scotland were contacted separately and directly. We purposively sampled for geographic spread and centre size, contacting centres by telephone and through

advertising on the NHFD. Twenty-four centres from across the UK agreed to take part.

## Data analysis

Descriptive statistics were used to provide frequency (counts, percentages) data concerning current services and practice, where the answer format provided predetermined response options (ZH). Where the response format was open ended, responses were coded and categorised into themes (MW). The integrated care pathways and physiotherapy exercise sheets returned to the team were qualitatively reviewed to provide description of commonalities and differences (MW).

## FOCUS GROUPS

Focus groups were completed at the three acute hospital sites across North Wales within Betsi Cadwaladr University Health Board. Three focus groups of members of the multidisciplinary rehabilitation teams in the community and the hospital, and four focus groups for patients and their carers were organised. Informed consent procedures were followed for recruitment, as approved by UK NHS North Wales Research Ethics Committee.

People who were over 65 years, were receiving rehabilitation following surgical repair of a proximal hip fracture within the last 3–12 months, were living independently prior to fracture and were able to provide informed consent were eligible to take part.

Eligible participants were identified from the NHFD, through the medical and nursing staff who were responsible for maintaining the database at each site.

## Data collection

Discussions were semistructured and run by a moderator (CAH) and comoderator (MW, NHW, JLR or NUD) using a topic guide containing open-ended questions regarding experiences, perceptions and beliefs about rehabilitation following proximal hip fracture. In the professionals' focus groups, patient scenarios were also used to stimulate discussion about the sort of rehabilitation patients would be likely to receive. In the later patient focus groups, we explored initial ideas for the intervention to gain feedback. The focus group discussions were digitally recorded and fully transcribed into the speaker's original language, with any portions in Welsh subsequently translated into English for analysis. The interviews were thematically analysed using the Framework approach.[69] The initial framework used was broadly developed from the theory areas identified as important to guide the realist review and it was used to index the transcripts. The researcher developed an initial interpretation of the data using the framework, and grouped the data into themes which were reviewed by a second researcher experienced in framework analysis. A third researcher reviewed the initial framework, original transcripts and the draft analysis to make final decisions on theme structure and content. The initial and third researchers agreed to the final analysis.

## Development of the intervention

The final programme theories and the results from the survey and the initial focus groups were discussed by all of the research teams in order to identify the important components of the rehabilitation intervention. The intervention components were discussed and refined in the final focus groups.

## RESULTS

A summary of the results is presented below. Further detail can be found in the final report to the funder.[18] The literature review, survey results and qualitative data were used to develop the following overarching working theory:

'In the context of patients with a great range and variety of pre-fracture physical and mental health co-morbidities affecting their ability to meet rehabilitation goals, a tailored intervention incorporating increased quality and amount of practice of exercise and activities of daily living in addition to usual rehabilitation leads to better confidence, mood, self-efficacy, function, mobility and reduced fear of falling.'

This was then broken down into three component programme theories described in figure 1.

### Programme theory 1: improve patient engagement by tailoring the intervention according to individual needs and preferences

Elderly patients with proximal hip fracture presenting with a range of prefracture physical and mental functioning and a variety of comorbidities need a rehabilitation programme that is tailored to individual needs in order to achieve appropriate outcomes such as improved physical functioning, greater mobility, reduced disability and independent living.

Findings from the realist review indicated that tailoring of patient care requires a detailed assessment of patients' prefracture level of functioning,[70 71] current cognitive status[72] and other comorbid conditions.[73 74] It should also involve collaborative decision-making through discussion and agreement with patients, their family and carers regarding: realistic and achievable,[74] but modifiable,[75] short-term and longer term goals of rehabilitation,[76 77] the most appropriate setting for rehabilitation suited to patients' needs and abilities,[78–80] and any adaptation of the physical environment to facilitate day-to-day activities.[80 81] In addition, the provision of enhanced support through active engagement of carers and rehabilitation professionals to motivate and facilitate the regular practice of exercises and activities of daily living,[62 63] improve health perceptions,[82] address and adjust outcome expectations,[63 83] and address information needs.[63 64 84]

In the survey, tailored rehabilitation was also identified as an important aspect of service provision. Respondents from all categories reported that the frequency of rehabilitation visits received was influenced by

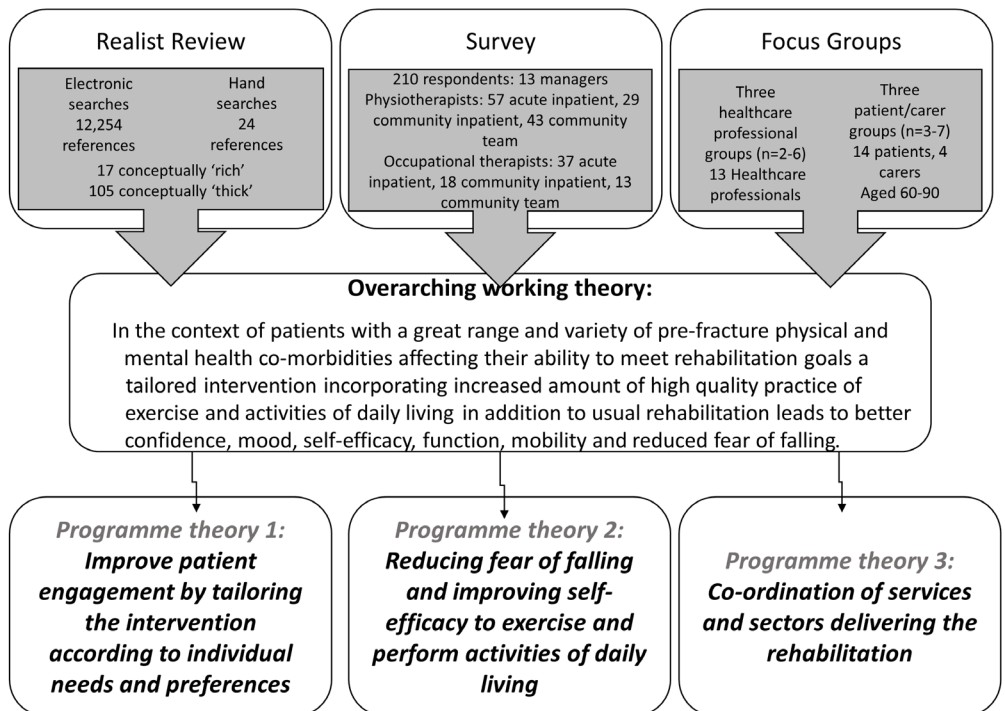

**Figure 1** Development of programme theories for informing the content of the enhanced rehabilitation intervention.

individual patient need (figure 2). Survey findings revealed that routine clinical practice was broadly in line with current guidance, but variability existed in the provision of services, especially in the community. Variation was reported in the frequency of rehabilitation visits following hip fracture (figure 2), with some services performing multiple visits a day and others visiting patients less than once a week. There was also variation in the types of activities included.

For example, while the majority of occupational therapists (over 95%) included prescribing equipment and practising activities of daily living in their usual activities, less than 50% included anxiety management and developing self-awareness. The importance of tailored care was also highlighted in focus groups (table 2), with many patients finding it hard to engage in strengthening exercises if they were not part of an individualised plan that focused on personal goals. The first

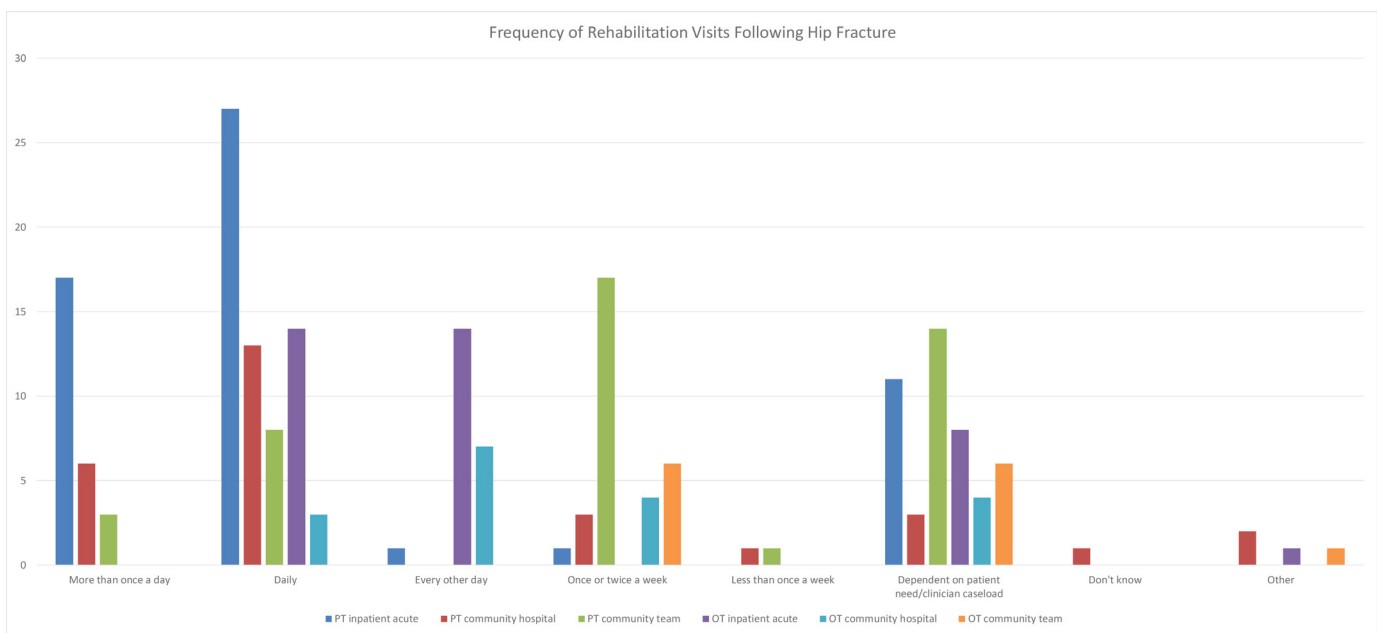

**Figure 2** Frequency of rehabilitation visits following hip fracture. OT, occupational therapy; PT, physiotherapy.

emergent theme from the focus groups related to the variability of care provision, which was partly because of individual tailoring of treatment, but also geographical variation in the availability of resources. Furthermore, comorbidities and prefracture functioning determined what patients were able to do and affected their attitude to exercise, which could be taken into account through individual tailoring of care plans.

### Programme theory 2: reducing fear of falling and improving self-efficacy to exercise and perform activities of daily living

Proximal hip fracture results in poor physical functioning, fear of falling, low mood and lack of self-efficacy requiring improved quality and increased amount of practice of physical exercises, activities of daily living and psychological tasks in order to gain mastery and control to improve confidence, mobility and physical functioning.

Enhancing the practice and quality of exercise and activities of daily living has both physical and psychological components.[63 64] This consists of supervision and coaching by health professionals in order to improve skills and confidence to promote independent and unsupervised practice,[80] with resulting increases in the duration, frequency and quality of exercises for improving strength, balance, gait and activities of daily living. Addressing psychological concerns is also important, particularly to improve mood.[59 85] Motivation to practise can be improved by setting appropriate, realistic goals and developing mechanisms for monitoring and providing feedback.[85 86]

According to the survey, more rehabilitation staff resources were needed to provide this support. Although patients' cognitive status, mood, self-efficacy and fear of falling were assessed, routine assessments using validated tools were not performed everywhere and the frequency that progress was assessed varied (table 3). The importance of psychological factors was also highlighted in focus groups where a second emergent theme was facilitators and barriers to rehabilitation, one of which was the reliance on patient's own self-motivation to seek out and access services. The level of patient engagement in the rehabilitation programme depended upon its perceived relevance to their day-to-day activities, and in the absence of this the amount of practice was likely to decline. A third focus group theme was the psychosocial effects of the fracture, fear of falling in particular, which reduced confidence and increased the reliance on walking aids. This fear

| Table 2 | Focus group themes and supporting quotes from patients, carers and healthcare professionals |
|---|---|
| **Theme** | **Supporting quotes** |
| Variation of rehabilitation care provision | *'It depends completely on the patient you can't just say well this is what is going to happen to every patient, they vary so much… there is different avenues depending on what they present.'* R4, community hospital physiotherapist, FG1121<br>*'It depends on what, what procedure she [the surgeon] has done to fix the fractured NOF [neck of femur] as to what level of interventions we do.'* R2, occupational therapist, FG1321<br>*'You are dealing with very angry relatives who were under the presumption that because they are under our service, that they will automatically get care and they won't, not unless there is a need.'* R1, clinical specialist physiotherapist, FG1321 |
| Facilitators and barriers to rehabilitation | *'There's a limit to what you can do at home, I got to the stage where I needed equipment… the first time I went to the gym and saw the physio there, I thought yes…It hurt, it was painful, but at least I felt I'm sure I'm going to get somewhere, and it has it's been brilliant.'* F2, female patient, FG1111<br>*'We refer a lot [to falls group], as long as they can get transport.'* R, FG1221<br>*'Seeing the physio, it's a mixture of more exercises and going through it but also it's the ability just to have someone to talk through things like what to do with the pain.'* M1, male patient, FG1111 |
| Psychosocial impact of hip fracture | *'Couple of women recently and have taken ages, whereas initially talking to them they are women you know sort of retired but really active, do loads, but then they have fallen and really I think it's more, you know the shock of the falling over and not being able to do things it does take them quite a long time to get over it.'* R3, acute hospital occupational therapist, FG1121<br>*"You think you are going to fall all the time, erm… so it is just practice I think, just keep doing it, keep doing little bits and erm…I had the reassurance from the physiotherapist who said 'no, by next summer you will be doing exactly what you were doing last summer'."* R1, female patient, FG1212<br>*'It's to do with personal care as well, and to raise confidence as well, that's a lot to do with it because people who have had the falls, it's their confidence really that's taken a big knock.'* R3, reablement team, FG1221 |
| Need for information | *'I didn't know what to do I didn't know whether to sit, and rest or try to exercise or what nobody told me anything… people don't explain… tell you so that you can understand. You just, left to ponder it over for yourself.'* R3, female patient, FG1211<br>*'There was a whole series of questions I had that had come up over the previous three weeks and I think the ability to go and talk to someone, with different experience and knowledge was very important for me now.'* M1, male patient, FG1111<br>*'Care is good, communication is rubbish.'* I3, male carer, FG1311 |

**Table 3** Validated measures reported to be used by physiotherapists and occupational therapists in acute and community settings

| Assessment | Acute hospital<br>Tools used | Community hospital<br>Tools used | Community team<br>Tools used |
|---|---|---|---|
| Cognitive status | Abbreviated Mental Test Score (AMTS)<br>Mini-Mental State Examination (MMSE)<br>Rapid Assessment Test for Delirium and Cognitive Impairment (4AT)<br>Six-Item Cognitive Impairment Test (6CIT)<br>Montreal Cognitive Assessment (MoCA)<br>Addenbrooke's Cognitive Examination (ACE-R) | 6CIT<br>MoCA<br>AMTS<br>MMSE<br>Middlesex Elderly Assessment of Mental State (MEAMS) | 6CIT<br>ACE-R<br>MMSE<br>MEAMS<br>MoCA<br>Cognitive (COG)Test<br>Rowland Universal Dementia Assessment Scale (RUDAS) |
| Mood | Hospital Anxiety and Depression Scale (HADS)<br>Geriatric Depression Scale (GDS)<br>Unified Assessment Proforma<br>MoCA | GDS<br>HADS<br>12-Item Short Form Health Survey (SF-12)<br>Canadian Occupational Performance Measure (COPM) | 6CIT<br>GDS<br>HADS<br>Therapy Outcome Measure (TOM) |
| Self-efficacy | Unified Assessment Proforma<br>10-Meter Walk | Falls Efficacy Scale (FES)<br>VAS<br>COPM | EQ-5D<br>TOM |
| Fear of falling | Visual Analogue Scale (VAS)<br>Berg Balance Scale<br>Elderly Mobility Scale<br>Oxford Hip Scale<br>Tinetti Assessment Tool | Falls Risk Assessment Tool (FRAT)<br>VAS<br>FES<br>Falls Efficacy Scale International (FES-I)<br>Tinetti Assessment Tool<br>COPM | FRAT<br>VAS |
| Health utility | | Euro Qol (EQ-5D) | EQ-5D |

affected engagement in the rehabilitation programme and impacted on wider social interactions, leading to feelings of isolation.

### Programme theory 3: coordination of services and sectors delivering the rehabilitation

The diversity of services provided by different disciplines, across sectors from a variety of funders requires a coordinated provision of the multidisciplinary rehabilitation programme in order to deliver appropriate physical, functional and psychological interventions to patients in a timely manner.

The coordination of multidisciplinary care from the acute hospital into the community required good communication between rehabilitation professionals and careful discharge planning. Patients valued the help and support they received from healthcare teams during their recovery and regarded this as the single most important factor in their recovery, so the provision of consistent and reliable care was vital.

Most respondents in the survey from both acute and community hospital settings reported that routine clinical practice was following the latest NICE (2011)[8] and SIGN (Scottish Intercollegiate Guidelines Network; 2009)[54] guidance. Multidisciplinary teams working with common goals across settings were a strength, but there was variability in service provision, especially with regard to what

was available in the community. Liaison between the acute hospital and the community could be improved, as could communication with patients and carers.

The fourth focus group theme was a need for more information for patients and their carers about what to expect following the hip fracture and how to access all of the available resources. The complexity in programme provision and the often poor communication between different sectors meant that rehabilitation was neither smooth nor seamless, and because of this lack of consistency, patients felt unsupported in their recovery. Patients and their carers required reassurance from qualified professionals about which activities were safe to perform in order to overcome these barriers, highlighting the role of the therapist as a mediator to improve their self-efficacy.

### Designing a rehabilitation intervention

Considering these findings the rehabilitation intervention needed to:
▶ identify individual goals with help from a therapist;
▶ enhance self-efficacy;
▶ increase the opportunity to practise prescribed exercises and activities of daily living;
▶ support the self-monitoring of progress towards identified goals;
▶ give encouragement and support from professionals;

- ▶ provide information on what to expect during rehabilitation;
- ▶ provide reliable and consistent care;
- ▶ signpost to other available services.

In order to address these, we developed a rehabilitation programme comprising both physical and psychological components (figure 3). The physical component consisted of additional rehabilitation sessions, tailored to individual need, following discharge home. The psychological component consisted of a patient-held information workbook, developed using an existing stroke rehabilitation workbook[87 88] as an exemplar, and a goal-setting diary. These aimed to improve patient engagement in the rehabilitation programme by giving patients a sense of ownership of their own recovery.

The additional sessions were also an opportunity for patients to obtain reassurance and guidance from a qualified healthcare professional. Similarly, the outcome of the psychological components aimed to increase confidence and self-efficacy that would affect patient's ability and willingness to perform exercises, thus improving their physical outcomes.

A detailed logic model of the intervention activities, their proposed long and short-term goals and how these target different components of the International Classification of Functioning framework has previously been published,[89] along with how the intervention addresses specific areas of existing NICE guidance for hip fracture rehabilitation.

## DISCUSSION

There were three programme theories from the realist review: improving patient engagement by tailoring the intervention according to individual needs and preferences; reducing fear of falling and improving self-efficacy to exercise and perform activities of daily living; and coordination of services and sectors delivering the rehabilitation. These were reflected in the survey data highlighting that while routine clinical practice was broadly in line with current guidance, there was variability in the provision of services, especially in the community, and that important psychological mediators such as self-efficacy and fear of falling were not routinely assessed using validated tools. They also agreed with the four focus group themes of: variation in rehabilitation care provided; the need for more information; facilitators and barriers to rehabilitation; and the psychosocial impact of hip fracture. These findings informed the development of a community-based rehabilitation intervention consisting of a psychological component delivered using a workbook and a patient-held goal-setting diary and a physical component comprising additional rehabilitation sessions.

Other studies have acknowledged the benefits of using realist review in intervention development,[90] but such methods have not previously been used in hip fracture rehabilitation research. This paper adds to the understanding of how a realist review can be used in conjunction with other methods to develop complex interventions which link individual intervention components with underlying programme theories.

The findings from the individual work packages are supported by existing literature. A qualitative study of physiotherapists' perceptions of rehabilitation also showed that tailoring of care to patient's individual needs, based on their own goals and level of support available, was an important component of successful rehabilitation.[91] A previous qualitative study exploring mobility levels pre and postfracture also reported that fear of falling, lack of confidence and reliance on others had an impact on patients' experiences of rehabilitation.[92] This study highlighted the need to include psychological components in rehabilitation interventions, supporting our findings about the importance of improving self-efficacy and confidence in patients with hip fracture. Our finding concerning patients' need for information from healthcare professionals and its importance in successful rehabilitation has also been previously identified.[93] A study into the challenges of team working in the rehabilitation of patients with hip fracture found that there were breakdowns in communication within multidisciplinary teams and issues relating to the organisation of resources and services, which led to variation in patient care.[94] Our intervention aims to address this by coordinating care through the means of a patient-held goal-setting diary.

This was the first realist review of rehabilitation following hip fracture and the first UK-wide survey aiming to describe rehabilitation for patients following hip fracture across acute and community settings since the introduction of NICE recommendations for rehabilitation in 2011.[5] As a realist review rather than a systematic review was performed, we did not attempt to summarise all of the evidence and judge whether rehabilitation programmes were effective, but rather sought to build an explanatory account of mechanisms behind rehabilitation. While a good range of respondents were sampled in the survey, it was not possible to sample settings, therapists and community service managers proportionately, which may impact on how representative findings are of the whole UK. Similarly, focus groups' findings relate specifically to the location we recruited from as this was the proposed setting for the delivery of the enhanced rehabilitation programme. We had also hoped to purposively sample patients with different levels of disability who had received different types of rehabilitation; however, it was not possible to identify these criteria from electronic medical records. Participants had a range of ages and experiences across the groups, though we were unable to sample those who were living independently prior to hip fracture, but who now lived in residential or nursing care, and those with cognitive impairment.

### Implications for future research and practice

Important implications for practice are the routine assessment of psychological variables and the inclusion

of psychological components in rehabilitation interventions. This study demonstrated the potential benefits of using a realist approach to complex intervention development and how a realist review could be used in conjunction with other established methods to provide an evidence base for a hip fracture rehabilitation intervention. This approach may be beneficial for developing complex interventions in other clinical areas and can be used to provide theories of how specific intervention components will facilitate their intended outcomes. The next phase in the MRC framework for evaluating complex interventions[12] was to test the feasibility of methods for a future trial of the developed intervention by testing its acceptability in a phase II feasibility study.[89 95]

**Author affiliations**
[1]School of Healthcare Sciences, Bangor University, Bangor, Gwynedd, UK
[2]Warwick Clinical Trials Unit, University of Warwick, Coventry, UK
[3]School of Psychology, Bangor University, Bangor, Gwynedd, UK
[4]Betsi Cadwaladr University Health Board, Bangor, Gwynedd, UK
[5]School of Sports, Health and Exercise Science, Bangor University, Bangor, Gwynedd, UK
[6]Division of Health and Social Care Research, King's College, London, UK
[7]School of Medicine, University of Nottingham, Nottingham, UK

**Acknowledgements** We would like to thank the patients, carers and staff who took part in the focus groups and survey, especially Glenys Mutter who maintains the Hip Fracture Database at Ysbyty Glan Clwyd and assisted us with identifying potential focus group participants. We would also like to thank: Dr Julia Hiscock for assistance with focus groups and consultation on qualitative analysis; Aaron Pritchard for consultation on qualitative analysis; Barbara France for running the electronic literature searches and study document retrieval; Annie Hendry for help in extracting the data from the included studies; Dr Richa Sinha for helping during reference screening and conceptual categorisation of potentially relevant studies for the realist review; Patricia Masterson Allgar and Professor Christopher Burton who were participants in the realist review workshops; John Spalding/ Bangor University library staff for helping in electronically inaccessible document retrieval through interlibrary loans.

**Contributors** NHW: Chief investigator (CI) responsible for study design, conduct and analysis, led intervention development, led writing of manuscript and is the guarantor; JLR: conducted focus group analysis and contributed to survey analysis, led writing of manuscript, development of intervention materials; NUD: conducted realist review, contributed to writing of manuscript; MW: conducted survey analysis, contributed to writing of manuscript; CAH: trial management, input to study and survey design, oversight of intervention development, conducted focus groups and survey, initial focus group analysis; JMC: assisted with realist review; ZH: input to study design, design of survey and initial survey analysis; VM: Co-investigator (Co-I) responsible for study design, provided health psychology expertise and methodological oversight; SA: Co-I, consultant orthogeriatrician, provided orthogeriatric expertise and input on intervention design; AL: Co-I responsible for study design, provided methodological oversight; CS and PL: Co-I contributing to methodology and study design, provided physiotherapy and rehabilitation expertise and input on intervention design; CW: Co-I contributing to study design and methodology; JRM: Co-I providing realist review expertise and methodological input. All authors were involved in writing and reviewing of the manuscript and decisions on final content.

**Funding** This research was funded by the Health Technology Assessment (HTA) programme of the National Institute for Health Research (NIHR) (HTA reference 11/33/03). The views and opinions expressed therein are those of the authors and do not necessarily reflect those of the HTA programme, NIHR, the National Health Service, or the Department of Health.

**Competing interests** None declared.

**Ethics approval** NHS North Wales Research Ethics Committee (West Ref 12/WA/0355).

**Provenance and peer review** Not commissioned; externally peer reviewed.

**Data sharing statement** The data sets generated and analysed during the current study are available from the corresponding author on reasonable request.

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
