## [Reviewer comments · BMJ Open]

ARTICLE DETAILS

TITLE (PROVISIONAL)	Development of an evidence-based complex intervention for community rehabilitation of hip fracture patients using realist review, survey and focus groups
AUTHORS	Roberts, Jessica; Din, Nafees; Williams, Michelle; Hawkes, Claire; Charles, Joanna; Hoare, Zoe; Morrison, Val; Alexander, Swapna; Lemmey, Andrew; Sackley, Catherine; Logan, Phillipa; Wilkinson, Clare; Rycroft-Malone, Jo; Williams, Nefyn

VERSION 1 – REVIEW

REVIEWER	Katherine Harding Eastern Health, Australia
REVIEW RETURNED	18-Oct-2016

GENERAL COMMENTS	This paper describes the development of an intervention for patient following hip fracture using a realist approach, incorporating a literature review, survey and focus groups. The paper has the potential to make a useful contribution to the literature in this area. Existing programs for hip fracture rehabilitation are often intensive and multi-disciplinary, making them expensive to deliver. However, as pointed out by the authors' outcomes for these patients are still often poor, with high rates of mortality and morbidity. Programs are often focussed on physical recovery, while other factors that are known to impact on return to activity, such as confidence and fear of falls, go unaddressed. My main concern with this paper is that in attempting to describe in detail all of the facets of data collection, the paper has become long and at times convoluted making it difficult to keep the reader engaged. The paper would benefit from "tightening up" throughout the manuscript, reducing words wherever possible, ensuring subheadings are clear and maintaining focus on key messages. The relationship between this paper and other work by the authors also adds another layer of potential confusion in some places. It is fine to reference the follow up trial, but try not to let that distract from the key messages of the current paper. Further specific points are outlined below. Introduction: Please ensure that all acronyms are spelled out in the first instance (eg MRC) and keep their use to a minimum. If an acronym is only
---

used once or twice, it is easier for the reader if it is spelled out in full. Final sentence of intro – “Has now been tested...” implies this work has been completed (which becomes clear in the discussion), but the reference is for a protocol. Also, as per the general point above, it is important to keep the focus of this paper on the development of the intervention and not let readers get distracted by the trial.

Therefore rather than ending the introduction with this reference (making it very prominent) Perhaps it would be better to move this reference to earlier in the paragraph, so that the development of the feasibility trial does not distract from the aim of the current paper. For example, modify the opening sentence of the paragraph to read “.....with this forming the basis for the development of our own evidence-based intervention for subsequent testing in a feasibility trial.[15]”

Study objectives – Several of these points are repetitive of information provided in the body of the introduction or in the methods. I suggest that this section be deleted and either the points incorporated in the text, or moved to a separate table or box.

Methods

Sections under the subheadings “Realist review” and “Developing Programme Theories” could be reduced and combined under a single subheading, providing a broad overview of the method. I suggest that this is then followed with sections under the four subheadings “Literature Search”, “Survey of UK Hip Fracture Centres”, “Focus Groups” and then “Development of the intervention”. Within each of the first three headings I recommend reducing detail within each section and avoiding the use of further subheadings. For example, under the survey section it would be sufficient to say that “Centres from England, Scotland and Wales were identified through publicly available information and advertising on the NHFD. Purposive sampling was used for geographic spread and centre size” without all the additional detail in paragraph 1, page 11.

The data analysis section of the survey method describes using frequency data, coding of qualitative responses and content analysis of exercise sheets. However none of this data is reported directly, and it is not clear how this data informed the development of the intervention.

Focus groups – once again given that there are three different methods of data collection in this study, brevity in each of these methods section is required. Inclusion criteria could be included in a single sentence rather than dot points (eg “Patients were included if they were over 65 years, were receiving rehabilitation following surgical repair of a proximal hip fracture within the last 3 to 12 months, were living independently prior to the fracture and able to provide informed consent”)

Ethical approval – was ethical approval provided for the entire study (ie survey as well as focus groups?) If so it would be better to have an overarching statement about ethical approval at the beginning of the methods section.

Analysis & credibility section – this section can also be substantially reduced. The key message of the paper is about bringing together the three methods of data collection as a whole and I feel that the paper is at times “bogged down” in the detail of the individual methods. If this was a stand alone qualitative paper describing the focus groups, this level of detail would be appropriate. But combined with a literature review and survey the reader needs an overview of each component, and then a clear description of how these three strands come together to inform the intervention.

Development of the Intervention – This fourth subheading under the methods should describe how the three sources of data were brought together to inform the intervention. This is, in my view, the most important section of the methods and if anything, this section could provide a little more detail about the process of integrating the data to design the intervention.

Results

The opening sentence of the results could be better worded to help readers to understand the structure of the sections that follow. For example:

“The literature review, survey results and qualitative data were used to develop three broad programme theories for consideration in an intervention for hip fracture rehabilitation. These are summarised in Figure 1.”

I would delete the subheading “Developing the program theories” and instead go straight into the subheading “Programme Theory 1....”

For each of the three Programme Theory sections, the authors essentially describe the findings from the literature review, survey and focus groups in turn, and how these findings support the theory. This structure works reasonably well, but through each of these sections I would like to see some (brief) reporting of the quantitative data from the survey.

So for example, in the sentence:

“Survey findings revealed that routine clinical practice was broadly in line with current guidance, but variability existed in the provision of services, especially in the community, and that psychological mediators including self-efficacy and fear of falling were not routinely assessed using validated tools.”

It would be good to incorporate some data into this sentence that provides some specific insights into the survey findings. For example, what percentage of respondents reported using validated tools? What was the nature of the variability described?

Incorporating some specific survey findings to support the programme theory development would be useful right throughout the results section.

It would also be useful to include some survey data and focus group data in tables to highlight key findings from each of these aspects of the data collection. This would provide useful information without having to add further text. A table of survey data could include some information to describe the sample, as well as quantitative findings from key questions. A separate table or figure could be used to illustrate key themes from the focus groups.

Discussion

The opening paragraph of the discussion is somewhat repetitive of the results section. Themes from the focus groups are first mentioned in the discussion – these need to be in the results, or in a separate table referred to from the results section.

The second paragraph of the discussion focuses primarily on limitations. While these should be acknowledged, it would be preferable to move this section later in the discussion.

The third paragraph makes some attempt to relate the current paper to previous literature. This is normally the main focus of a discussion and I would suggest moving it forward. It is not entirely clear what point the authors are making in this paragraph – it opens with a comment on realist review, and then goes on to a series of statements of findings of other papers. I think that there is scope to improve the discussion by breaking up this paragraph, and using these references to highlight more clearly how the current paper and

	relates to previous findings in relation to confidence, communication, tailoring of interventions and teamwork. Implications for further research: I would delete the second half of this paragraph relating to the next phase of the work – if anything this could be included as a stand alone paragraph earlier in the discussion. Finishing the paper with this information distracts from the main message about using a realist approach to intervention development.
--	---

REVIEWER	Paula M. van Wyk Assistant Professor University of Windsor Canada
REVIEW RETURNED	17-Dec-2016

GENERAL COMMENTS	I would like to know why this paper is being submitted for publication following that of the phase II paper for FEMuR (reference [15], as well as the protocol paper). This does not make efficient or feasible sense. There is a claim in the paper that there is a paucity of evidence based interventions in the literature. However, you are moving forward with your intervention before the evidence behind your intervention is even submitted for publication? That seems circumspect. This paper also claims that there will be theory development. I do not see theory development anywhere in this paper. There are statements made, but there is no actual theorizing. Even with figure one, I appreciate the illustration, and I am not even arguing against what is written - just that this does not present itself to me as a theory. Please refer to work by Per Nilsen, or other publications in Implementation Science that discuss theories, models and frameworks. I do not mean to sound insulting, as I am aware that some of the authors, mainly Rycroft-Malone have made instrumental advancements with respect to what is considered evidence in evidence-based practice. Rather, in the current state, I am failing to see how theory or theories are actually being presented. Perhaps the paper needs to focus solely on the evidence behind the intervention being developed as that is a stronger point than claiming theories are also being developed. Furthermore, the results of the paper are not new. They have been found in numerous studies in the recent years. Thus, I am left wondering, other than a realist approach, what is this paper actually adding to the literature? Thus, I encourage the authors to rethink their submission to make a more powerful impact on the literature, or it is just repetitive of what already exists. With the participants, why were there 129 physiotherapists in comparison to 68 occupational therapists surveyed? Later in the manuscript the authors note that one finding was more robust among the physiotherapists. That seems commonsense since they were a large majority of the participation population. This may also be a limitation you want to address. I have an issue as well with the authors using value laden language. There has been a strong movement towards removing the labels
--

with respect to patient populations. The authors even address that interventions need to be more tailored to the patients, yet they continuously label them first as a hip fracture rather than a person. They are not hip fracture patients, they are patients who experienced a hip fracture.

Why are only proximal femoral fractures focused on? Why not all hip fractures in general?

Sentence two on page four is also poorly worded. Please revise.

Line 49 - 56 that begins "Whilst there have been many systematic reviews.." please provide references.

Furthermore, the data collection for this study was conducted in 2013. It is currently December 2016. There have been many advancements in the literature in the last 3-4 years, thus making this work outdated.

Which databases were used, stating 21 is not enough information.

Please provide further details as to what is meant by "rich", "thick", and "thin".

How many separate reviewers were involved? Who was involved with resolving discrepancies (e.g. was an additional reviewer brought in, was it just among this who did the screening..)

Please be consistent with terminology. At times the authors state older adults, other places elderly, etc... Same with the word healthcare, it is sometimes written as two words. Please amend.

For the data collection section on page 10, the authors note that there were three different versions of the survey. The only difference that was noted was that the therapists were subdivided by setting. I have no opposition to the different versions. However, it is currently unclear how they differed and whether this had implications for analyzing and comparing the results.

For box 1 - "time limited", what were the parameters of this?

Line 37-39 on page 13, "agreed the final analysis" - I believe there may be a word missing here.

Again, with respect to reference [15], the first approximately 20 words are the exact same as this publication. Please ensure that these two papers are not reusing sentences as that is not appropriate.

I do believe that there is validity in this work; it should not be ignored or outright rejected. However, I do believe some major revisions are required in order to advance this paper to a stage where a true impact in the literature can be made. Thus, I truly hope the authors take this as an opportunity to revise their current submission.

REVIEWER	Julia McGregor Nursing Midwifery & Allied Health Professions Research Unit University of Stirling Scotland
REVIEW RETURNED	20-Jan-2017

GENERAL COMMENTS	Abstract: The abstract summarises the study and its relationship to the next phase of research (a feasibility study). It would perhaps be helpful to have a short statement about the background/context to the study here (ie that hip fracture rehabilitation guidance for clinicians lacks a strong evidence base for effectiveness). I would also have liked to see a brief explanation in the abstract as to why realist review was chosen as a method - understanding the real-world contexts, mechanisms and outcomes of effective interventions - as this would be of interest to clinicians & researchers. The study design is strengthened by the use of expert and professional feedback, descriptive data about practice applications and qualitative methods. The use of theoretical discussion in patient focus groups strengthens the focus on patient experience and expertise. I wondered if activity data for Scotland was available from another source? I would like to have seen a brief explanation of the use of the framework approach, so that readers understand the choice of this technique The study highlights potential barriers and facilitators for interventions based on the identified mechanisms – the need for more resources – which is relevant to economic considerations for policymakers and managers It would be good to see some more of the data – data tables with the descriptive statistics, also some examples of the qualitative data particularly from patients, to illustrate their experiences and views on intervention development Some more detail on the framework analysis would also be helpful in understanding how the theory explains the context-mechanism-outcome relationships. I appreciate there may be a word limit issue with including more detail, but perhaps this could be provided as supplementary information? I enjoyed this paper which conveyed a strong sense of the importance for clinical practice of understanding real-world barriers and facilitators of recommended interventions, particularly those which directly influence patient ability to benefit from interventions
--

REVIEWER	Dr Margaret A Coulter Smith Queen Margaret University Edinburgh Scotland United Kingdom
REVIEW RETURNED	21-Feb-2017

GENERAL COMMENTS	Title- This clearly explains the focus of the paper. Abstract- The last claim in the conclusions could be reduced- perhaps better to qualify this with ...may offer advantages ... Last sentence- 'testing' of rehab intervention not addressed in this paper- to be reported in another paper.
---

Article summary-

Strengths and limitations

'The MRC framework was used to develop a complex intervention for hip fracture rehabilitation ...' for accuracy the authors could reduce the claim to state that the work undertaken aligns with stage one of the MRC framework?

'The methods used to develop this rehabilitation programme are applicable....' Perhaps rephrase to 'may be applicable...' and is similar to the way this is reported towards the end of the paper.

Background

Overall this section reads very well.

Suggestions for inclusion:

Could report UK statistics on annual incidence of hip fracture in the over 65s as authors report 12 month mortality, percentage who regain function within 12 months, and the economic costs.

It would be helpful to include a definition of rehabilitation for the purposes of this paper- a narrow/ focused view of rehabilitation seems to underpin the body of the paper- for example there is less attention to management of chronic conditions such as osteoporosis which directly impacts on bone health and increases the risk of fracture, guidance on smoking cessation, diet, nutrition, use of supplements, osteoporosis medications, types of exercise that promote bone health and other lifestyle factors. Does the time-period for rehabilitation extend from hospital discharge to one year post fracture? E.g. Content in Box 1 'Working definition of multidisciplinary rehabilitation' is currently focused on strength (muscle strength?), confidence and activities of daily living rather than also including a focus on promoting bone health.

The authors rightly indicate the MRC framework does not prescribe how interventions should be developed. At this point the authors could also refer to some of the options such as Charles Abraham's on Behaviour Change Theory and Intervention Mapping Framework, or approaches to modelling and optimizing complex intervention using factorial designs as discussed by Walter Sermeus in Richards & Rahm Hallberg (2015), or perhaps the potential use of Delphi Studies for this purpose. Levati et al. 2016 have published a useful scoping review focusing on the optimisation of complex health interventions prior to a RCT in Pilot and Feasibility Studies (2016) 2:17 DOI 10.1186/s40814-016-0058-y

Methods

Realist review

To what extent were the authors able to 'establish which components produced the reported outcomes for specific patient groups in which circumstances' given the limitations they acknowledge later in the paper?

Developing programme theories

Could define term 'theoretically sound primary studies'.

Feedback from experts in health psychology, rehabilitation, implementation research was combined with initial survey and focus group findings and intermediate programme theories to build ... context, mechanisms, outcome for final programme theories. May need to state how contributions from the various sources were managed as some groups had limited representation e.g. patients?

Study objectives-

These are broad but are in line with the intended focus of the realist review.

Is 2) fully achievable with the methodology used- reliance on secondary reports 'to find out what is being provided'.

Box 1- Could define term 'frail elderly'.

Literature search

Screening and categorisation of references- could be more specific here- age ranges for 'elderly adults', location, time period, stage in patient journey post hip fracture.

Discrepancies between reviewers were 'resolved after discussion'. This could be illustrated with some examples.

Definition of terms 'conceptually rich', 'thick' or 'thin'- the difference between conceptually rich and conceptually thick might benefit from further explanation on P.8.

Build and 'test' theories (is 'test' the best descriptor for the process used- perhaps examine works better here?) How were the 'most relevant concepts' decided upon- was this by the agreement of all team members and did the concepts link to the definition of multidisciplinary rehabilitation reported earlier in Box 1?

'Testing' the theories with quantitative and qualitative evidence

Refining the theories in the light of quantitative and qualitative evidence...

Conceptually rich and conceptually thick- might these categories ever overlap? Could be useful to illustrate with an example of each?

Survey of UK Hip Fracture Centres

Define 'hip fracture centre'. Are these the centres where hip fracture repair surgery is undertaken?

Strong bias towards physiotherapists compared to others in the MDT (may reflect a narrow/ focused definition of rehabilitation in this context).

Could state inclusion and exclusion criteria for volunteer sample used. Are the authors able to estimate the total population from which the sample was drawn?

Data collection- content of questionnaire tool? Whilst Scotland data are not in the NHFD they are reported in the SHFD and ISD and are accessible to all.

P.13. Good reporting of quality criteria guiding the qualitative interviews and data analysis.

Results

Programme Theory 1 and 2 sections read well.

Page 17 Programme Theory 3 particularly important- i.e. 'help and support received from the health care team is vital to recovery'.

Authors refer to their earlier publication of a Logic model of the intervention activities which is very useful for readers.

p. 21- line 52 typo 'its'

References- Typo at reference 10, line 27 'Rameses..'

Both figures are labelled appropriately and report clear content.

REVIEWER	Geoff Wong University of Oxford, United Kingdom
REVIEW RETURNED	07-Mar-2017

GENERAL COMMENTS	Specific comments: Abstract page 2 lines 12 to 25: This section of the abstract does not mention anything about the the realist review processes. Please would you include something about the realist review and also some details about the interview processes. I appreciate that the word count allowance for abstracts may limit how much you might be able to write. Page 4 lines 29 to 31: When you say an evidence based intervention has not been published, are there however evidence based interventions that are in existence? Page 5 lines 15 to 20: I was interested in your concept of "intermediate theories". Are these an intermediate step in the way you move from a broad and 'untestable' programme theory to one or more theories, specified in the middle-range and realist way (i.e. the CMO configuration)? Or are these theories something else? Page 6 lines 19 to 51: You have listed study objectives, but do you have any research questions? Please would you add these to the manuscript. Page 7 lines 24 to 27: Please would you describe how you decided if a primary study was theoretically sound. Page 7 line 24: You undertook a "scoping search". Please would you provide more details on the processes you used for this. Who did it, how etc. etc. Such details would be of interest to readers who might wish to use your methodology. Page 7 lines 39 to 41: You mention members of the "evaluation team" and "evaluation workshops". I thought you were doing a realist review. Is this a typo? Also, who are the members of this team? Page 7 lines 44 to 54: You mention that you combined the scoping search data with the survey and focus groups. Does that mean you undertook the survey and focus groups at the same time or at some other point in time? It would be very helpful for readers if you would please provide a figure or diagram of the stages within your project.
--

Also, may I please ask if you deliberately added details about your data analysis processes here as opposed to later in the manuscript where you mention data extraction and analysis?
If so, why?
Having read further along the manuscript, it would help if you provided more details on what these data extraction forms were like and if they were piloted and refined.

Page 8 Literature search section

Please would you provide more details on the literature search processes. Who did the searching, how, were there any checks for consistency? Which 21 databases were searched - were these the same ones as the NICE guideline review?
How did you adapt the NICE review? Why did you do these various adaptations?
Did you pilot and refine your review?
Was your review adaptations guided by what you had found in the scoping review?
Did you only undertake one search? How was the searching informed (if at all) by your programme theory and/or the scoping review.
Etc. etc.

Page 8 Screening and categorisation of references section:

In this section, you have a PICO, but please would you provide details of your inclusion and exclusion criteria. Or are they in Box 1?

More details are also needed on the screening processes.

When you mention "Separate reviewers ..." was there two or three or more?
Did they work independently or in some other way?
What were they screening? Titles, abstracts, keywords, all, something else?
What happened if there was no agreement after discussion.
Who and how was the decision made on what constituted 'rich', 'thick' or 'thin'?
What role, if any, did the CMOs you developed from the scoping review play in these decisions?

You mention the word "study" more than once. Does this mean that only studies were included or were other document types eligible for inclusion?
If you had only included studies, why was this done?

Finally to clarify, when you mention that the purpose "... was to make the database .. concepts." by this, did you mean you focused your data extraction and analysis back from richest or thickest to the least rich or thin?

Page 9 Data extraction section section.

Please would you provide more details on the checking for accuracy process.

I note you used the MMAT to assess study quality. Why did you rate studies/documents using this tool?

Why not use Pawson's concept of rigour?

How does this related to your categorisation of studies as 'thick' or 'thin' - if at all)?

What did you do with the results of the MMAT when it came to inclusion and data analysis?

IF you had included non-study documents, how would you rate these?

Page 10: Testing theories section

Please would you provide some clarification and more detail on your data analysis process.

Were you developing a programme theory for each type of rehabilitation intervention? Is it these programme theories you were comparing and contrasting to look for patterns?

Also what kinds of patterns? Of outcomes under certain contexts or something else?

How and where do the contexts, mechanism and outcomes relate to your programme theories?

Did you then go on to develop a more abstract but still middle range realist programme theory for post op rehab? Or did you just refine the programme theories for each intervention type into a more realist version?

Page 10: Survey development

Please could you clarify if the surveys were piloted and refined prior to use.

As you are developing a programme theory, did you use this programme theory to inform the contents of your survey?

If not why not?

Page 10: Data collection section:

Please would you provide a description of the survey process.

Was this a paper survey?

Were reminders sent?

etc.

Page 11: Data analysis section

Did you use any software programmes to help you with your analysis.

Other than frequencies did you undertake any further analyses?

Who did the analysis and were any quality control processes used?

How (if at all) were these data related to and analysed along with the programme theory from the realist review?

Why did you undertake thematic analysis and not try to analyse these data using the same realist logic of analysis used in the programme theory?

It does not seem to me that what you have done is what you mentioned as the title of this section on page 10 line 3 - "Testing the theories ...".

This seems more like a stand alone section of research at present.

Page 12 Focus groups section

Please would you provide details on who ran the focus groups.

How did you select the hospital site and why?

Page 12 Focus groups inclusion criteria for patients.

Please would you provide more details on the inclusion and exclusion criteria for patients AND health care professionals.

Page 12 Data collection section.

Was your topic guide and patient scenarios related to or informed by the programme theory from the realist review.

If not, why not?

At present these focus groups sound like a separate project that does not somehow relate to the realist review and survey.

Page 13: Analysis, credibility and plausibility section.

I am very confused here.

How does thematic analysis using a framework relate to the realist logic of analysis set out by Pawson and Tilley, CMO configurations and programme theory (if at all)?

If they do, an illustrative example of how the data analyses that involves data from the realist review, surveys and focus groups should be provided to aid the reader in understanding the analytic processes used.

This might be focused on a section of the programme theory and could be provided as a supplementary file if preferred - either in the methods or results section.

Such an example would be of great help to those interested in the methodology you have used to develop your intervention.

Page 13 Development of the intervention

Again as for the section above on Analysis, plausibility and credibility, it would be helpful if you provide an illustrative example.

Page 14 onwards: Results section.

A lot more detail is needed in this section on the results of the various methods used. Of course I realise that you have provided some detail in Figure 1, but a bit more is needed.

For the realist review, I would have expected details on the search processes, screening processes, document characteristics, a PRISMA style flow diagram of disposition of studies etc.

I would strongly suggest you consult the RAMESES publication guidelines for realist syntheses for more details.

The same would go for the surveys and focus groups - more basic details needed for the findings section.

Of course I realise word count can be an issue, so these may be provided as supplementary files if preferred.

Page 14 Programme theory 1 (Box).

What you have conceptualised as functioning as a mechanism in this box is not a mechanism. It is an intervention strategy (i.e. something that you do).

There is also very limited dis-aggregation of the CMOs.

There is more than one context listed here, one 'mechanism' and at least 4 outcomes.

It's hard to tell what you need to change context-wise to get the outcomes you want and also what the causal mechanism(s) are.

I would suggestion you look at the following resources to see if you might be able to re-analyse the data so that a realist logic of analysis has clearly applied:

- RAMESES Training materials

http://www.ramesesproject.org/media/Realist_reviews_training_materials.pdf

- Astbury B.,Leeuw F. Unpacking Black Boxes: Mechanisms and Theory Building in Evaluation. American Journal of Evaluation 2010;31:363-81.

- Dalkin S, Greenhalgh J, Jones D, Cunningham B, Lhussier M. What's in a mechanism? Development of a key concept in realist evaluation.

Implementation Science 2015;10.

Page 14 line 41 to Page 15 line 44:

What would help me in terms of transparency would be of you would please provide for each Programme theory, the data you have used to develop your arguments that underpin the CMOs within your programme theory.

At present the data is quite descriptive and it is hard to see how the data in this section relates to the Programme theory above.

Page 16. Programme theory 2.

The comments I have made for Programme theory 1 above applies to Programme theory 2 as well.

Page 17. Programme theory 3.

The comments I have made for Programme theory 1 above applies to Programme theory 3 as well.

Page 19 Designing a rehabilitation intervention.

My comments from Page 13 (Development of the intervention) applies to this section.

I found it hard to follow from your interpretations and arguments from data to programme theory. You then also mention a a logic model in this section. How does the logic model relate to the programme theories above and your 'Overarching theory' from Figure 1?

I note for example that patients will be given a "patient-held information workbook". However I was not clear where the data was in your realist review, focus groups and/or surveys to explain why this workbook would 'work' better than anything else for some patients vs other patients or in what contexts.

It is this kind of detail analysis that I would have expected to see in your analysis.

Page 19 Discussion section.

You may wish to reconsider revising the Discussion section in light of my comments on the methods and result sections above as well as the RAMESES publication standards for realist syntheses:

<http://bmcmedicine.biomedcentral.com/articles/10.1186/1741-7015-11-21>

Page 30 Figure 1

This is an excellent diagram that provides a lot of information very clearly in a highly accessible way.

I have a number of observations of this diagram:

a) You might wish to call you "Overarching working theory" your "Initial or rough programme theory".

The programme theories below are then sub-components of programme theory.

b) What you have assigned as functioning as mechanism in the "Overarching working theory" box are not mechanisms - they are intervention strategies. In other words, things that you would want to do to change contexts so that correct mechanisms are triggered to cause the multiple outcomes you have listed.

REVIEWER	Joanne Greenhalgh School of Sociology and Social Policy
REVIEW RETURNED	20-Apr-2017

GENERAL COMMENTS	The paper is ambitious (perhaps too ambitious) in its scope. I wasn't quite certain whether the paper was focused on presenting findings or explaining/explicating a methodology. The title suggests the latter - it suggests that the paper is going to explain how realist methods were used to develop a complex intervention. However, my overall feeling at the end of the paper is that there was more focus on providing information about the individual trees and much less on explaining how they work together to form a wood. In other words, much of the methods section of the paper focuses on explaining how each separate data set was assembled and analysed but there is much less discussion of how and through what processes all these data were integrated through realist logic to underpin intervention development. The results section does attempt this integration, but as a reader I felt that the potential richness of each data set disappeared and the realist logic underpinning the integration of these data got a bit lost. I would have liked to have seen more detail regarding the initial programme theories and how each data set has contributed to testing and refining these. Its not quite clear whether the theories represent 'refined theories' - ie an 'end product' or 'theories that then require further testing using realist logic', ie for example, when intervention is further evaluated. I appreciate the challenge the authors (I imagine) have faced in writing up their work because it is difficult to both provide sufficient detail about the individual pieces of data collection at the same time as provide a more 'meta-view' of how these individual pieces have been integrated. I wonder therefore whether there is merit in the idea of either allowing the authors more words to do this, or perhaps putting some of the detail regarding data collection and analysis in a box/supplementary file along with the manuscript. The more radical alternative is to suggest writing up a realist review in a separate paper and then in a further paper explain how its findings were tested and refined via focus groups and placed in the context of the findings of the survey to develop an intervention. However, I do think with some revision and restructuring it is possible achieve this within a single paper. I have made some comments on specific sections of the paper in the hope the authors might find them helpful in revising the paper:  1. In the methods section, it would be helpful to have a section which explains how each strand of data collection related to the other and how they each informed the process of theory testing and refinement. It would be useful to know what the role of each strand of data collection was - it seems to me that the survey was not really used for theory testing but to provide information on current context - so how was this integrated with the review and focus groups? The diagram (Figure 1) does not really show the dynamics of this process. For example, its not quite clear whether the three different strands were conducted independently and the integration process occurred only at the end (if so - how was this done?) or whether integration occurred throughout the process, around the process of theory gleaning, testing and refining, in keeping with realist logic. 2. When each item of data collection is described, I would have liked to have seen a sentence of two which explains how data collection
---

	and analysis was used to glean, test or consolidate theories. This is especially the case for the focus groups. To use Manzano's (2016) typology - were the focus groups about gleaning, testing or refining theory - or all three? How did the focus groups relate to the realist review - were they conducted after or alongside the review and how did the findings of the review inform the conduct of the focus groups? The section on in the methods on 'development of the intervention' needs expanding to give more detail in the integration process and the role of theory in this process. 3. What is the status of the three programme theories? As discussed above, are they an 'end product' or do they represent a 'mid point' in the theory testing process - ie hypotheses that can then be tested when the intervention is further evaluated? 4. I don't want to come over as the 'CMO police' but my overall sense when reading the PT, as presented in the boxes, is that they focus much more on delineating the 'resources' element of mechanisms and much less on the 'responses' aspect. There is more detail in the narrative text surrounding each, but I did want to ask (for example, in relation to PT 1)- so what is it about tailoring that enables these desired outcomes to occur? Maybe it is just about signposting this more clearly - for example, the sentence that really seemed to come close to grasping the 'response' element was the findings from the focus groups that indicated patients found it hard to engage with generalised goals and preferred individualised ones - this begs the intriguing question of - why? Furthermore, the explication of context seemed to represent the 'existing context' - ie the problem that the intervention is trying to solve, rather than hypothesising how possible variations in context might support or constrain the different mechanisms through which rehabilitation is expected to work. This is fine and its often how PT theories start, but it does suggest the further testing will be needed to/result in a more detailed understanding how variations in context might shape the way the intervention works. Again, would be worth signposting this. 5. In the final section of the results, Figure 2 does a good job of connecting the programme theories and the intervention itself. It would be helpful if these connections were made more clearly in the text narrative itself.
--	---

VERSION 1 – AUTHOR RESPONSE

Reviewer: 1

Reviewer Name: Katherine Harding

Institution and Country: Eastern Health, Australia

Please state any competing interests: None declared

Please leave your comments for the authors below

This paper describes the development of an intervention for patient following hip fracture using a realist approach, incorporating a literature review, survey and focus groups. The paper has the potential to make a useful contribution to the literature in this area. Existing programs for hip fracture rehabilitation are often intensive and multi-disciplinary, making them expensive to deliver. However, as pointed out by the authors' outcomes for these patients are still often poor, with high rates of mortality and morbidity. Programs are often focussed on physical recovery, while other factors that are known to impact on return to activity, such as confidence and fear of falls, go unaddressed.

My main concern with this paper is that in attempting to describe in detail all of the facets of data collection, the paper has become long and at times convoluted making it difficult to keep the reader

engaged. The paper would benefit from “tightening up” throughout the manuscript, reducing words wherever possible, ensuring subheadings are clear and maintaining focus on key messages. The relationship between this paper and other work by the authors also adds another layer of potential confusion in some places. It is fine to reference the follow up trial, but try not to let that distract from the key messages of the current paper.

Further specific points are outlined below.

REPLY: Thank you. We have clarified the text and clarified the relationship with the feasibility study.

Introduction:

Please ensure that all acronyms are spelled out in the first instance (eg MRC) and keep their use to a minimum. If an acronym is only used once or twice, it is easier for the reader if it is spelled out in full.

Reply: Thank you, we have amended this.

Final sentence of intro – “Has now been tested...” implies this work has been completed (which becomes clear in the discussion), but the reference is for a protocol. Also, as per the general point above, it is important to keep the focus of this paper on the development of the intervention and not let readers get distracted by the trial. Therefore rather than ending the introduction with this reference (making it very prominent) Perhaps it would be better to move this reference to earlier in the paragraph, so that the development of the feasibility trial does not distract from the aim of the current paper. For example, modify the opening sentence of the paragraph to read “.....with this forming the basis for the development of our own evidence-based intervention for subsequent testing in a feasibility trial.[15]”

Reply: We have amended this as suggested for clarity.

Study objectives – Several of these points are repetitive of information provided in the body of the introduction or in the methods. I suggest that this section be deleted and either the points incorporated in the text, or moved to a separate table or box.

Reply: Thank you, we have deleted this section as we agree it duplicates much of the information in the final paragraph of the introduction

Methods

Sections under the subheadings “Realist review” and “Developing Programme Theories” could be reduced and combined under a single subheading, providing a broad overview of the method. I suggest that this is then followed with sections under the four subheadings “Literature Search”, “Survey of UK Hip Fracture Centres”, “Focus Groups” and then “Development of the intervention”. Within each of the first three headings I recommend reducing detail within each section and avoiding the use of further subheadings. For example, under the survey section it would be sufficient to say that “Centres from England, Scotland and Wales were identified through publicly available information and advertising on the NHFD. Purposive sampling was used for geographic spread and centre size” without all the additional detail in paragraph 1, page 11.

Reply: Thank you for the detailed suggestions. We have clarified the process and streamlined this section where possible. However, we have had to balance this with requests from other reviewers for additional detail.

The data analysis section of the survey method describes using frequency data, coding of qualitative responses and content analysis of exercise sheets. However none of this data is reported directly,

and it is not clear how this data informed the development of the intervention.

Reply: We have added additional figures to report some of the pertinent survey and focus group data.

Focus groups – once again given that there are three different methods of data collection in this study, brevity in each of these methods section is required. Inclusion criteria could be included in a single sentence rather than dot points (eg “Patients were included if they were over 65 years, were receiving rehabilitation following surgical repair of a proximal hip fracture within the last 3 to 12 months, were living independently prior to the fracture and able to provide informed consent”)

Reply: Thank you, we have amended this accordingly.

Ethical approval – was ethical approval provided for the entire study (ie survey as well as focus groups?) If so it would be better to have an overarching statement about ethical approval at the beginning of the methods section.

Reply: We were advised by Betsi Cadwaladr University Health Board Research and Development department that ethical approval was only required for the focus groups.

Analysis & credibility section – this section can also be substantially reduced. The key message of the paper is about bringing together the three methods of data collection as a whole and I feel that the paper is at times “bogged down” in the detail of the individual methods. If this was a stand alone qualitative paper describing the focus groups, this level of detail would be appropriate. But combined with a literature review and survey the reader needs an overview of each component, and then a clear description of how these three strands come together to inform the intervention.

Reply: Whilst balancing requests from other reviewers for more detail regarding Framework Analysis, we have amended and streamlined this section accordingly.

Development of the Intervention – This fourth subheading under the methods should describe how the three sources of data were brought together to inform the intervention. This is, in my view, the most important section of the methods and if anything, this section could provide a little more detail about the process of integrating the data to design the intervention.

Reply: Thank you, we have extended this section and clarified how the three separate methods were used in conjunction to develop the intervention.

Results

The opening sentence of the results could be better worded to help readers to understand the structure of the sections that follow. For example:

“The literature review, survey results and qualitative data were used to develop three broad programme theories for consideration in an intervention for hip fracture rehabilitation. These are summarised in Figure 1.”

I would delete the subheading “Developing the program theories” and instead go straight into the subheading “Programme Theory 1....”

Reply: Thank you, we have amended this section and expanded on your suggestions to clarify the overarching theory.

For each of the three Programme Theory sections, the authors essentially describe the findings from the literature review, survey and focus groups in turn, and how these findings support the theory. This structure works reasonably well, but through each of these sections I would like to see some (brief)

reporting of the quantitative data from the survey.

So for example, in the sentence:

“Survey findings revealed that routine clinical practice was broadly in line with current guidance, but variability existed in the provision of services, especially in the community, and that psychological mediators including self-efficacy and fear of falling were not routinely assessed using validated tools.” It would be good to incorporate some data into this sentence that provides some specific insights into the survey findings. For example, what percentage of respondents reported using validated tools? What was the nature of the variability described? Incorporating some specific survey findings to support the programme theory development would be useful right throughout the results section.

It would also be useful to include some survey data and focus group data in tables to highlight key findings from each of these aspects of the data collection. This would provide useful information without having to add further text. A table of survey data could include some information to describe the sample, as well as quantitative findings from key questions.

Reply: We have expanded on survey findings in the text to support the development of the programme theories. A figure has been added to demonstrate the variability in frequency of visits across the different groups. We have also included a table to demonstrate the different variety of validated measures used, particularly with reference to psychological measures and self-efficacy, which is pertinent to the development of programme theory two.

A separate table or figure could be used to illustrate key themes from the focus groups.

Reply: Thank you, a table illustrating the focus group themes, with supporting quotes, has now been added.

Discussion

The opening paragraph of the discussion is somewhat repetitive of the results section. Themes from the focus groups are first mentioned in the discussion – these need to be in the results, or in a separate table referred to from the results section.

Reply: We have added further detail of the focus group results to the results section

The second paragraph of the discussion focuses primarily on limitations. While these should be acknowledged, it would be preferable to move this section later in the discussion.

Reply: Thank you, we have moved this section accordingly.

The third paragraph makes some attempt to relate the current paper to previous literature. This is normally the main focus of a discussion and I would suggest moving it forward. It is not entirely clear what point the authors are making in this paragraph – it opens with a comment on realist review, and then goes on to a series of statements of findings of other papers. I think that there is scope to improve the discussion by breaking up this paragraph, and using these references to highlight more clearly how the current paper and relates to previous findings in relation to confidence, communication, tailoring of interventions and teamwork.

Reply: We have expanded on our initial comments to clarify how our paper contributes to this and fits in with the existing literature.

Implications for further research:

I would delete the second half of this paragraph relating to the next phase of the work – if anything

this could be included as a stand alone paragraph earlier in the discussion. Finishing the paper with this information distracts from the main message about using a realist approach to intervention development.

Reply: We have amended this section to maintain the focus on the findings of this paper.

Reviewer: 2

Reviewer Name: Paula M. van Wyk

Institution and Country: Assistant Professor, University of Windsor, Canada

Please state any competing interests: None declared.

Please leave your comments for the authors below

I would like to thank the authors for their submission that took a realist approach for the development of a rehabilitation intervention for patients who experienced a hip fracture.

This is the first paper that I have read or reviewed that has taken a realist approach. I commend you on that.

However, that is perhaps the only novel aspect of this paper, and that presents some concerns regarding publication. There were also some limitations that were not addressed, that I think need to be. With revision, I believe this paper can be strengthened and improved which would be a benefit for the journal, the authors and the readership. In no way do I intend any insult or offense with my review, this is strictly my opinion.

I would like to know why this paper is being submitted for publication following that of the phase II paper for FEMuR (reference [15], as well as the protocol paper). This does not make efficient or feasible sense. There is a claim in the paper that there is a paucity of evidence based interventions in the literature. However, you are moving forward with your intervention before the evidence behind your intervention is even submitted for publication? That seems circumspect.

This paper also claims that there will be theory development. I do not see theory development anywhere in this paper. There are statements made, but there is no actual theorizing. Even with figure one, I appreciate the illustration, and I am not even arguing against what is written - just that this does not present itself to me as a theory. Please refer to work by Per Nilsen, or other publications in Implementation Science that discuss theories, models and frameworks. I do not mean to sound insulting, as I am aware that some of the authors, mainly Rycroft-Malone have made instrumental advancements with respect to what is considered evidence in evidence-based practice. Rather, in the current state, I am failing to see how theory or theories are actually being presented. Perhaps the paper needs to focus solely on the evidence behind the intervention being developed as that is a stronger point than claiming theories are also being developed.

Furthermore, the results of the paper are not new. They have been found in numerous studies in the recent years. Thus, I am left wondering, other than a realist approach, what is this paper actually adding to the literature?

Thus, I encourage the authors to rethink their submission to make a more powerful impact on the literature, or it is just repetitive of what already exists.

Reply: Thank you for your feedback on the paper, your comments have been very helpful in making our substantial revisions. We have amended the title of the paper to clarify that the focus of the paper is on the development of a complex intervention for hip fracture rehabilitation, with a realist review contributing to the development of this intervention. Our aim was to describe and clarify our processes in developing a pragmatic intervention and the relevance of the individual components of this

intervention with relation to the evidence for them. We hope that the amendments we have made have addressed the issues raised here.

With the participants, why were there 129 physiotherapists in comparison to 68 occupational therapists surveyed? Later in the manuscript the authors note that one finding was more robust among the physiotherapists. That seems commonsense since they were a large majority of the participation population. This may also be a limitation you want to address.

Reply: Thank you, this comment related to the proportions of each therapist group who indicated they based the frequency of their visits on patient need rather than a direct comparison of numbers between the groups. We have deleted this comment and added further detail and a figure to clarify.

I have an issue as well with the authors using value laden language. There has been a strong movement towards removing the labels with respect to patient populations. The authors even address that interventions need to be more tailored to the patients, yet they continuously label them first as a hip fracture rather than a person. They are not hip fracture patients, they are patients who experienced a hip fracture.

Why are only proximal femoral fractures focused on? Why not all hip fractures in general?

Reply: The HTA commissioned brief related specifically to proximal femoral fracture. All other hip fractures were therefore excluded.

Sentence two on page four is also poorly worded. Please revise.

Reply: This has been done.

Line 49 - 56 that begins "Whilst there have been many systematic reviews.." please provide references.

Reply: References have been added

Furthermore, the data collection for this study was conducted in 2013. It is currently December 2016. There have been many advancements in the literature in the last 3-4 years, thus making this work outdated.

Reply: Whilst we appreciate there has been further literature published since the review was conducted we are only proposing to report on how our findings were used to develop our intervention.

Which databases were used, stating 21 is not enough information.

Reply: These have been added in an appendix.

Please provide further details as to what is meant by "rich", "thick", and "thin".

Reply: Due to the word limit of the manuscript, and balancing requests from other reviewers for less detail on individual methods, we have been unable to expand on the definitions of these categories. However, references are in place to papers which give detail on these definitions.

How many separate reviewers were involved? Who was involved with resolving discrepancies (e.g.

was an additional reviewer brought in, was it just among this who did the screening..)

Reply: This has been detailed throughout the manuscript

Please be consistent with terminology. At times the authors state older adults, other places elderly, etc... Same with the word healthcare, it is sometimes written as two words. Please amend.

Reply: Thank you, this has now been amended to provide consistency.

For the data collection section on page 10, the authors note that there were three different versions of the survey. The only difference that was noted was that the therapists were subdivided by setting. I have no opposition to the different versions. However, it is currently unclear how they differed and whether this had implications for analyzing and comparing the results.

Reply: This has been expanded, and further detail is available in our final HTA report.

For box 1 - "time limited", what were the parameters of this?

Reply: Further detail has been added.

Line 37-39 on page 13, "agreed the final analysis" - I believe there may be a word missing here.

Again, with respect to reference [15], the first approximately 20 words are the exact same as this publication. Please ensure that these two papers are not reusing sentences as that is not appropriate.

Reply: Thank you, This has been re-written.

I do believe that there is validity in this work; it should not be ignored or outright rejected. However, I do believe some major revisions are required in order to advance this paper to a stage where a true impact in the literature can be made. Thus, I truly hope the authors take this as an opportunity to revise their current submission.

Reviewer: 3

Reviewer Name: Julia McGregor

Institution and Country: Nursing Midwifery & Allied Health Professions Research Unit, University of Stirling, Scotland

Please state any competing interests: None declared

Please leave your comments for the authors below

I found this a very interesting paper which addresses the issue of identifying evidence for the mechanisms which might facilitate and strengthen the effectiveness of interventions for hip fracture rehabilitation, and uses the evidence to develop a complex intervention to improve outcomes. It is likely to be of interest, and important, to a range of health professionals and allied health professionals involved with decisions about care and treatment of the elderly. It is very clearly written. As someone with extensive practice experience I appreciated the accessible explanation of the realist approach and the focus on clinical application - I feel this is a very accessible paper for practitioners.

Reply: Thank you very much for your feedback.

Abstract: The abstract summarises the study and its relationship to the next phase of research (a feasibility study). It would perhaps be helpful to have a short statement about the background/context to the study here (ie that hip fracture rehabilitation guidance for clinicians lacks a strong evidence base for effectiveness). I would also have liked to see a brief explanation in the abstract as to why realist review was chosen as a method - understanding the real-world contexts, mechanisms and outcomes of effective interventions - as this would be of interest to clinicians & researchers.

Reply: Thank you, the abstract has been amended and your comments incorporated where possible within the restrictions of the word count and advised structure.

The study design is strengthened by the use of expert and professional feedback, descriptive data about practice applications and qualitative methods. The use of theoretical discussion in patient focus groups strengthens the focus on patient experience and expertise.

I wondered if activity data for Scotland was available from another source?

Reply: Following feedback from other reviewers relating to the level of detail for each methodology, this section has now been removed. However, there were centres from Scotland included in the survey, only activity data was missing. We hope our amendments have clarified any confusion regarding this.

I would like to have seen a brief explanation of the use of the framework approach, so that readers understand the choice of this technique

Reply: We are limited by the word count but have included references to further information on Framework Analysis. Further detail on focus group results have also been added which we hope will aid understanding of the focus group work and how it contributed to the intervention development.

The study highlights potential barriers and facilitators for interventions based on the identified mechanisms – the need for more resources – which is relevant to economic considerations for policymakers and managers

It would be good to see some more of the data – data tables with the descriptive statistics, also some examples of the qualitative data particularly from patients, to illustrate their experiences and views on intervention development

Reply: Thank you, we have added further detail on the survey and focus group data in the text, figures and tables.

Some more detail on the framework analysis would also be helpful in understanding how the theory explains the context-mechanism-outcome relationships. I appreciate there may be a word limit issue with including more detail, but perhaps this could be provided as supplementary information?

I enjoyed this paper which conveyed a strong sense of the importance for clinical practice of understanding real-world barriers and facilitators of recommended interventions, particularly those which directly influence patient ability to benefit from interventions

Reviewer: 4

Please leave your comments for the authors below

This paper is well written and has a clear and logical structure. The realist approach to complex intervention development reported here makes a useful and interesting contribution to the literature on developing complex interventions within rehabilitation programmes for people post hip fracture with particular reference to a study population in Wales. The authors note the main research design limitations and generally do not overstate their claims- minor changes are suggested to ensure consistency in this aspect.

Reply: Thank you very much for your positive feedback and suggestions. We have made changes to ensure consistency and clarified the overall message of the paper.

Title- This clearly explains the focus of the paper.

REPLY: Thank you. This has been amended in line with the editorial comments.

Abstract-

The last claim in the conclusions could be reduced- perhaps better to qualify this with _may offer advantages _

REPLY: We agree. This has been amended.

Last sentence- 'testing' of rehab intervention not addressed in this paper- to be reported in another paper.

REPLY: This has been amended for clarity.

Article summary-

Strengths and limitations

'The MRC framework was used to develop a complex intervention for hip fracture rehabilitation' for accuracy the authors could reduce the claim to state that the work undertaken aligns with stage one of the MRC framework?

'The methods used to develop this rehabilitation programme are applicable_.' Perhaps rephrase to 'may be applicable_' and is similar to the way this is reported towards the end of the paper.

REPLY: We agree. These have been amended.

Background

Overall this section reads very well.

REPLY: Thank you

Suggestions for inclusion:

Could report UK statistics on annual incidence of hip fracture in the over 65s as authors report 12 month mortality, percentage who regain function within 12 months, and the economic costs.

REPLY: We have made some changes to the background, but limited word count precludes us from making major additions.

It would be helpful to include a definition of rehabilitation for the purposes of this paper- narrow/ focused view of rehabilitation seems to underpin the body of the paper- for example there is less attention to management of chronic conditions such as osteoporosis which directly impacts on bone health and increases the risk of fracture, guidance on smoking cessation, diet, nutrition, use of supplements, osteoporosis medications, types of exercise that promote bone health and other lifestyle factors. Does the time-period for rehabilitation extend from hospital discharge to one year post fracture? E.g. Content in Box I 'Working definition of multidisciplinary rehabilitation' is currently focused on strength (muscle strength?), confidence and activities of daily living rather than also

including a focus on promoting bone health.

REPLY: The management of chronic conditions such as osteoporosis is important and the improvement of bone health is part of standard NHS practice. Similarly, guidance on smoking cessation and good nutrition is part of best practice. We did not intend to replace standard practice but to augment it and as such we have focussed on multidisciplinary rehabilitation as defined in Box 1. We have added detail to box 1 to clarify the time period.

The authors rightly indicate the MRC framework does not prescribe how interventions should be developed. At this point the authors could also refer to some of the options such as Charles Abraham's on Behaviour Change Theory and Intervention Mapping Framework, or approaches to modelling and optimizing complex intervention using factorial designs as discussed by Walter Sermeus in Richards & Rahm Hallberg (2015), or perhaps the potential use of Delphi Studies for this purpose. Levati et al. 2016 have published a useful scoping review focusing on the optimisation of complex health interventions prior to a RCT in Pilot and Feasibility Studies (2016) 2:17 DOI 10.1186/s40814-016-0058-y

REPLY: Thank you for the references. We have described our approach in the methods section.

Methods

Realist review

To what extent were the authors able to 'establish which components produced the reported outcomes for specific patient groups in which circumstances' given the limitations they acknowledge later in the paper?

Developing programme theories

Could define term 'theoretically sound primary studies'.

REPLY: We have clarified this sentence by describing the studies as theoretically rich. The papers were categorised according to theoretical richness as defined by Ritzer 1991 and Roen 2006 (references 62 and 63).

Feedback from experts in health psychology, rehabilitation, implementation research was combined with initial survey and focus group findings and intermediate programme theories to build _ context, mechanisms, outcome for final programme theories. May need to state how contributions from the various sources were managed as some groups had limited representation e.g. patients?

REPLY: We have added further detail relating to intervention development, including clarification that feedback on initial intervention components was sought from focus group participants.

Study objectives-

These are broad but are in line with the intended focus of the realist review.

Is 2) fully achievable with the methodology used- reliance on secondary reports 'to find out what is being provided'.

REPLY: Thank you, this section has been removed and the objectives incorporated into the background section on advice of other reviewers.

Box 1- Could define term 'frail elderly'.

REPLY: Over 65 years old has been added for clarity.

Literature search

Screening and categorisation of references- could be more specific here- age ranges for 'elderly adults', location, time period, stage in patient journey post hip fracture.

REPLY: We have amended Box 1

Discrepancies between reviewers were 'resolved after discussion'. This could be illustrated with some examples.

REPLY: Thank you for your suggestion but we have been unable to add further detail because of word limits and request from other reviewers to restrict methodological detail.

Definition of terms 'conceptually rich', 'thick' or 'thin'- the difference between conceptually rich and conceptually thick might benefit from further explanation on P.8.

REPLY: We have added further detail of the method here. Definitions are available in the references provided.

Build and 'test' theories (is 'test' the best descriptor for the process used- perhaps examine works better here?) How were the 'most relevant concepts' decided upon- was this by the agreement of all team members and did the concepts link to the definition of multidisciplinary rehabilitation reported earlier in Box 1?

REPLY: Thank you, we have amended this. We have added further detail of team members involved in the processes.

'Testing' the theories with quantitative and qualitative evidence

Refining the theories in the light of quantitative and qualitative evidence_

Conceptually rich and conceptually thick- might these categories ever overlap? Could be useful to illustrate with an example of each?

REPLY: Due to word limit we are unable to add this level of detail here but it is available in our final HTA report which has now been referenced.

Survey of UK Hip Fracture Centres

Define 'hip fracture centre'. Are these the centres where hip fracture repair surgery is undertaken?

REPLY: Yes, we have clarified this in the text.

Strong bias towards physiotherapists compared to others in the MDT (may reflect a narrow/ focused definition of rehabilitation in this context).

Could state inclusion and exclusion criteria for volunteer sample used. Are the authors able to estimate the total population from which the sample was drawn?

REPLY: As there is no register or centrally held record of physiotherapists or OTs working in hip fracture rehabilitation, we were not able to establish the population of such professionals in the UK or use such a register as a sampling frame. We decided to advertise the survey on the websites of the chartered society of physiotherapists and college of OTs/British Association of OTs to target special interest groups where possible and on the NHTA website. WE also asked those who saw the advert

to pass the survey web link onto any colleagues working in this field. We also asked therapy service managers to pass the survey web link onto their therapy staff.

Data collection- content of questionnaire tool? Whilst Scotland data are not in the NHFD they are reported in the SHFD and ISD and are accessible to all.

P.13. Good reporting of quality criteria guiding the qualitative interviews and data analysis.

REPLY: We have added some detail relating to the content of the survey but are limited by word count.

Results

Programme Theory 1 and 2 sections read well.

Page 17 Programme Theory 3 particularly important- i.e. 'help and support received from the health care team is vital to recovery'. Authors refer to their earlier publication of a Logic model of the intervention activities which is very useful for readers.

p. 21- line 52 typo 'its'

References- Typo at reference 10, line 27 'Rameses..'

Both figures are labelled appropriately and report clear content.

Reviewer: 5

Reviewer Name: Geoff Wong

Institution and Country: University of Oxford, United Kingdom

Please state any competing interests: None declared

Please leave your comments for the authors below

Thank you for asking me to review this manuscript.

I read it with interest both from a clinical and methodological point of view.

Whilst as a clinically active GP I am aware of the issues around hip fractures and rehabilitation, I am no content expert on this topic area and so can only comment from a generalists point on view.

The main thrust of my comments relate to the realist research approach and other methods used in this manuscript.

Overall I felt that the manuscript covered an important area in health services research and used an interesting range of approaches to 'theorise up' the intervention. As a general comment, my feeling was that more detail was needed through the manuscript. Such additional detail would really add to the clarity and methodological usefulness of this manuscript.

REPLY: This paper aims to summarise the three methodological strands involved in the development of the rehabilitation intervention. A detailed account is in the HTA final report which is in press; Williams NH, Roberts JL, Din NU, Charles JM, Totton N, Williams M, Mawdesley K, Hawkes CA, Morrison V, Lemmey A, Edwards RhT, Hoare Z, Pritchard AW, Woods RT, Alexander S, Sackley C, Logan P, Wilkinson C, Rycroft-Malone J. A multidisciplinary rehabilitation package following hip fracture: Fracture in the Elderly Multidisciplinary Rehabilitation (FEMuR). Health Technol Assess 2017; 21 (in press). A separate realist review paper is in preparation. We have added some additional detail to improve the clarity and methodological usefulness of this paper.

Specific comments:

Abstract page 2 lines 12 to 25:

This section of the abstract does not mention anything about the the realist review processes. Please would you include something about the realist review and also some details about the interview processes.

I appreciate that the word count allowance for abstracts may limit how much you might be able to write.

REPLY: Additional detail added but restricted by the word count.

Page 4 lines 29 to 31:

When you say an evidence based intervention has not been published, are there however evidence based interventions that are in existence?

REPLY: This sentence has been shortened for clarity.

Page 5 lines 15 to 20:

I was interested in your concept of "intermediate theories".

Are these an intermediate step in the way you move from a broad and 'untestable' programme theory to one or more theories, specified in the middle-range and realist way (i.e. the CMO configuration)?

Or are these theories something else?

REPLY: Yes, these are intermediate steps to producing the final programme theories.

Page 6 lines 19 to 51:

You have listed study objectives, but do you have any research questions?

Please would you add these to the manuscript?

REPLY: The review research questions were as follows:-

1. What community based multidisciplinary rehabilitation programmes have been developed and what were their main aims (intended outcomes)?

2. What were the mechanisms by which community-based rehabilitation of hip fracture patients are believed to result in their intended outcomes?

3. What are the identified contexts that determine whether different mechanisms yield intended outcomes?

Given the evidence in response to 1-3 we also drew conclusions regarding:

4. In what circumstances are the rehabilitation programmes likely to be effective and cost-effective if implemented in the NHS?

5. In what circumstances, with which combination of mechanisms and contexts are the rehabilitation programmes likely to generate unintended effects or costs?

As this paper is a summary of the three methods used to develop the rehabilitation intervention, rather than a realist review paper, they have not been added to this paper.

Page 7 lines 24 to 27:

Please would you describe how you decided if a primary study was theoretically sound.

REPLY: This sentence has been changed to 'theoretically rich primary studies' to improve clarity.

Page 7 line 24:

You undertook a "scoping search".

Please would you provide more details on the processes you used for this. Who did it, how etc. etc.

Such details would be of interest to readers who might wish to use your methodology.

REPLY: Additional detail has been added. Further detail is available in the final report.

A scoping search of the literature was done in the Medline, EMBASE and PubMed databases for relevant systematic reviews concerning multidisciplinary rehabilitation following hip fracture, stroke and the frail elderly using the broad search terms 'rehabilitation', 'frail', 'elderly', 'stroke', 'hip/femur fracture.' These reviews and their reference lists of included studies were the starting point for identifying both the implicit and explicit theories behind the success or failure of rehabilitation programmes or their components. Existing UK and international guidelines were also searched for additional contributions to theory development.

Initial immersion in the rehabilitation literature sought to identify an initial list of relevant intermediate programme theories. We scanned relevant primary studies and other linked papers with a strong theoretical content identified from the reference list of these reviews. This process helped to map out important areas and research gaps in the literature, resulting in a list of unanswered questions under different domains related to receivers (patients), deliverers (health care and rehabilitation teams), programmes (rehabilitation) and settings or systems (hospital, community etc.) used to deliver such rehabilitation programmes.

These lists of questions were formulated into statements to signify how the different domains of a programme interact and might affect all the agencies (stakeholders) involved. These were termed 'intermediate programme theories' which were refined during discussions between members of the evaluation team and with other researchers engaged in similar realist evaluations (i.e. two realist evaluation workshops in the School of Health Care Sciences, Bangor University convened by one of the senior researchers, JR-M). To keep track of these emerging programme theories a table was constructed in which the theories could be recorded, cross-referenced and commented upon. Feedback from the workshops was integrated into this table.

This list enabled the building of context (C), mechanisms (M) and outcome (O) configurations that are the basis of developing the final programme theories of how complex programmes (systems) work in certain contexts to produce intended (or unintended) outcomes. The initial list of these CMO configurations was again refined iteratively in team meetings.

Page 7 lines 39 to 41:

You mention members of the "evaluation team" and "evaluation workshops".

I thought you were doing a realist review. Is this a typo?

Also, who are the members of this team?

REPLY: This has been changed to "research team" and "realist workshops". The team members have been added.

Page 7 lines 44 to 54:

You mention that you combined the scoping search data with the survey and focus groups.

Does that mean you undertook the survey and focus groups at the same time or at some other point in time?

REPLY: These were undertaken at the same time. Results from the survey of multidisciplinary rehabilitation programmes and focus groups of patients, carers and rehabilitation professionals were also used to refine these programme theories, which were incorporated into the review as it progressed. Findings from the health professional survey that contributed to theory development included the importance of tailoring, the importance of feedback mechanisms, and variation in the delivery of rehabilitation in different areas based on the availability of staff and facilities. The focus groups with patients and their carers highlighted unmet information needs about the process of recovery, the availability of services that patients were entitled to access but were not necessarily aware of, and geographic variation in the provision of services.

It would be very helpful for readers if you would please provide a figure or diagram of the stages within your project.

REPLY: As we have added further figures and tables to present survey and focus group data, we have clarified the stages of the intervention development and how they are connected in the text.

Also, may I please ask if you deliberately added details about your data analysis processes here as opposed to later in the manuscript where you mention data extraction and analysis?
If so, why?

REPLY: This section has been clarified. The summary of findings from our initial immersion in the literature, feedback from the above mentioned meetings and workshops, from experts in health psychology, rehabilitation and implementation research, and the findings of the patient focus groups and health professional survey were integrated into our candidate programme theories. The emergent list of intermediate working theories was used as the basis for bespoke data extraction forms.

Having read further along the manuscript, it would help if you provided more details on what these data extraction forms were like and if they were piloted and refined.

REPLY: Additional detail has been added. Further detail is available in the final report. Two sets of bespoke data extraction forms were developed on an Access database to be used in two stages to extract data from both comparative studies (randomised/ quasi/ non-randomised controlled trials, comparative cohort and case control studies) and non-comparative studies (qualitative involving patients or health professionals, service evaluations, routinely collected database studies). The data extraction form for comparative studies was designed to collect data from each study on: study characteristics (design, sample type, sample size); description of intervention and control; process details (fidelity of intervention, dosage); contextual factors in the study setting; outcomes collected; theories or mechanisms postulated by the authors to explain the results. The data extraction form for non-comparative studies was designed to collect data on study characteristics, research method, theoretical approach, sample type, description of intervention/programme, method of analysis, and evidence to test the programme theories.

The forms were used in two stages to extract the data from included studies and test the intermediate and final programme theories. The first set of forms was used to populate the initial themes with evidence from effective (or ineffective) components of rehabilitation programmes and how these interacted with outcomes in given contexts. These themes were then refined into statements which led to the development of intermediate programme theories. The second set of forms was used to test these theories and adjudicate between competing theories

Page 8 Literature search section

Please would you provide more details on the literature search processes.

Who did the searching, how, were there any checks for consistency?

Which 21 databases were searched - were these the same ones as the NICE guideline review?

How did you adapt the NICE review? Why did you do these various adaptations?

Did you pilot and refine your review?

Were your review adaptations guided by what you had found in the scoping review?

Did you only undertake one search? How was the searching informed (if at all) by your programme theory and/or the scoping review.

Etc. etc.

REPLY: Additional detail has been added. Further detail is available in the final report. The literature search strategy used by the NICE guideline review of multidisciplinary rehabilitation programmes for hip fracture was adapted to encompass all of the theory areas of the first phase of the review process. No filters for study design were applied so that all study designs such as randomised and non-randomised controlled trials, observational, economic, and qualitative studies could be included. The

following databases were searched for published, semi-published and grey literature from inception to Feb 2013:

- MEDLINE
- MEDLINE In-Process & Other Non-Indexed Citations
- OLDMEDLINE
- EMBASE
- Cumulative Index to Nursing and Allied Health Literature (CINAHL)
- Allied and Complimentary Medicine Database (AMED)
- British Nursing Index
- Health Management Information Consortium (HMIC)
- PsychINFO
- Cochrane Central Register of Controlled Trials (CENTRAL)
- Database of Abstracts of Reviews of Effects (DARE)
- Cochrane Database of Systematic Reviews (CDSR)
- Health Technology Assessment (HTA) Database
- NHS Economic Evaluation Database (NHS EED)
- Science Citation Index
- Social Science Citation Index (SSCI)
- Index to Scientific & Technical Proceedings (ISTP)
- Physiotherapy Evidence Database (PEDro)
- BIOSIS
- System for Information on Grey Literature In Europe (SIGLE)
- Web of Knowledge Index of Theses and Dissertations

These have been added in an appendix.

Identified references were de-duplicated and transferred to bibliographic software (Endnote) to facilitate assessment for inclusion and categorisation of relevant studies. Multiple publications arising from the same study were identified, grouped together and represented by a single reference. Realist review involves iterative and purposive literature searching, so citations were tracked (forwards and backwards) and internet search engines such as Google Scholar and individual publisher websites were used to identify additional evidence as the review progressed and as new ideas emerged. The reference lists of previous systematic reviews and included studies were also screened to identify relevant studies. Using this method, no attempt was made to include every relevant study but materials were retrieved purposively to answer specific questions or test specific theories. The process stopped when sufficient evidence had been collected to fulfil its purpose i.e. the question raised by the theory had been answered. Conversely, if a new question arose, it triggered further literature searching to answer the question posed and to determine its fit within existing theory or if a new theory needed to be formulated.

Page 8 Screening and categorisation of references section:

In this section, you have a PICO, but please would you provide details of your inclusion and exclusion criteria. Or are they in Box 1?

REPLY: Yes these are included in Box 1.

More details are also needed on the screening processes.

When you mention "Separate reviewers ..." was there two or three or more?

Did they work independently or in some other way?

What were they screening? Titles, abstracts, keywords, all, something else?

What happened if there was no agreement after discussion.

Who and how was the decision made on what constituted 'rich', 'thick' or 'thin'?

What role, if any, did the CMOs you developed from the scoping review play in these decisions?

REPLY: Additional detail has been added. Further detail is available in the final report. A working definition of multidisciplinary rehabilitation to be used for screening sources of evidence was adapted from a review of intermediate care services, the working definition of which was presented in box 1. This definition was used when screening the titles and abstracts of identified studies in the Endnote® library for relevance. This was done independently by two separate reviewers and discrepancies resolved after discussion. In addition these potentially relevant papers were categorised according to study type: systematic review, randomised and non-RCTs, observational studies, economic evaluations, qualitative studies. There were no language restrictions and non-English publications were translated whenever possible using Google Translate™ or by seeking help from other researcher colleagues in our department who could speak and translate other languages. Potentially relevant references were conceptually categorised as either 'rich', 'thick', or 'thin' based on the criteria described by Ritzer and Roen and as used in a recent review of intermediate care. This process made the database manageable and enabled the information to be gleaned from the most appropriate studies for theory building and testing. After the initial screening and conceptual categorisation of the references in the Endnote library, potentially relevant studies were exported into a separate library for full document retrieval. Study inclusion criteria were applied to these retrieved documents by two reviewers independently, and conflicts were resolved by mutual discussion or after consulting a third reviewer. A list of all studies to be included was prepared for data extraction.

You mention the word "study" more than once. Does this mean that only studies were included or were other document types eligible for inclusion? If you had only included studies, why was this done?

REPLY: Systematic reviews and guidelines were also included in the scoping review. The main searches included all types of studies that presented explicit theories about the success or failure of an intervention in certain contexts or had implicit information that could be used to confirm or refute a theory. Study designs included: RCTs, quasi-RCTs, non-RCTs, cohort studies (with concurrent or historical controls), case-control studies, before and after studies, full economic evaluations and qualitative studies.

Finally to clarify, when you mention that the purpose "... was to make the database .. concepts." by this, did you mean you focused your data extraction and analysis back from richest or thickest to the least rich or think?

Reply: Yes, this has been clarified in the text.

Page 9 Data extraction section section.

Please would you provide more details on the checking for accuracy process.

I note you used the MMAT to assess study quality. Why did you rate studies/documents using this tool?

Why not use Pawson's concept of rigour?

How does this related to your categorisation of studies as 'thick' or 'thin' - if at all)?

What did you do with the results of the MMAT when it came to inclusion and data analysis?

IF you had included non-study documents, how would you rate these?

REPLY: Data were extracted by one reviewer and checked for accuracy by a second. Inconsistencies or disagreements were resolved by mutual discussion and agreement after checking against the source study.

Study quality was assessed using the mixed methods appraisal tool (MMAT) for mixed studies reviews, which can be used across different study designs (qualitative, trials, observational). The

purpose of appraising the 'quality' of studies was to assist in the judgment of relevance and rigour of different evidence from a 'fitness for purpose' perspective rather than scoring the studies for acceptance or rejection.

Page 10: Testing theories section

Please would you provide some clarification and more detail on you data analysis process.

Were you developing a programme theory for each type of rehabilitation intervention? Is it these programme theories you were comparing and contrasting to look for patterns?

Also what kinds of patterns? Of outcomes under certain contexts or something else?

How and where do the contexts, mechanism and outcomes relate to your programme theories?

Did you then go on to develop a more abstract but still middle range realist programme theory for post op rehab? Or did you just refine the programme theories for each intervention type into a more realist version?

REPLY: This section has been clarified. Further detail is available in the final report. Data for each individual study were examined in terms of the identified programme theories and the interaction between mechanisms, context and outcomes. Next, the data across the different studies were examined to detect patterns and themes for each theory in turn. Separate fields were created in the access database to capture these interactions as well as raw statement data from the included studies to support reviewers' reflections. Data synthesis involved individual reflection and team discussions to question the integrity of each theory, adjudicate between competing theories, consider the same theory in comparative settings, and compare the stated theory with actual practice. Where candidate theories failed to explain the data, new theories were sought from included studies or from the wider rehabilitation literature, such as rehabilitation following stroke or following in-patient admission after being unable to stand. Writing the narrative of the review was guided by the final theories that emerged from this process. The literature analysis relating to each of the identified theories was presented in detail in the final report, followed by a data summary to show the relationships between data themes and the theories in the final synthesis. Extracts were taken from the participant quotes (patient, carer or health professional) reported in the included qualitative studies, and were used as evidence to support subthemes under the main theory areas.

Page 10: Survey development

Please could you clarify if the surveys were piloted and refined prior to use.

As you are developing a programme theory, did you use this programme theory to inform the contents of your survey? If not why not?

REPLY: The following has been added; "The questionnaires were piloted on members of staff across one health board in Wales, and a few minor amendments were made." The survey was performed alongside the realist review before the final programme theories had been developed. The survey findings did however contribute to the final programme theories, including the importance of tailoring, the importance of feedback mechanisms and variation in delivery.

Page 10: Data collection section:

Please would you provide a description of the survey process.

Was this a paper survey?

Were reminders sent? etc.

REPLY: The survey was web-based and this has been included in the manuscript. We have had to balance the request from other reviewers for less detail on methodology, but further detail is provided in the final report. The sample of acute centres including Scotland was achieved by contacting acute hospitals by telephone and email. We advertised the survey of physiotherapists and occupational therapists through their professional bodies and on the National Hip Fracture Database website. We

also asked those who saw it to pass the link on to any colleagues they knew working in this field. We also asked therapy service managers completing their survey to pass the survey web link on to their therapy staff.

Page 11: Data analysis section

Did you use any software programmes to help you with your analysis.

Other than frequencies did you undertake any further analyses?

Who did the analysis and were any quality control processes used?

REPLY: Descriptive statistics only performed by ZH, the trial statistician, which has been added.

How (if at all) were these data related to and analysed along with the programme theory from the realist review? Why did you undertake thematic analysis and not try to analyse these data using the same realist logic of analysis used in the programme theory?

REPLY: Programme theories were used to draft the initial framework for the analysis, and this has been clarified in the text. Our intention was to combine the use of realist review with traditional survey and focus group methodologies to develop a complex intervention. We have made clarifications and additions to the paper which we hope have made this clearer.

It does not seem to me that what you have done is what you mentioned as the title of this section on page 10 line 3 - "Testing the theories ...". This seems more like a stand alone section of research at present.

REPLY: The data analysis section refers to the survey. The section heading has been italicized to make this clearer. "Testing the theories" is in the previous realist review section. The findings contributed to the final programme theories as mentioned in the development of the intervention section.

Page 12 Focus groups section

Please would you provide details on who ran the focus groups.

REPLY: This has been added.

How did you select the hospital site and why?

REPLY: These were the sites where we carried out a feasibility study in Phase II of the study.

Page 12 Focus groups inclusion criteria for patients.

Please would you provide more details on the inclusion and exclusion criteria for patients AND health care professionals.

REPLY: Healthcare professionals were recruited from members of the multidisciplinary rehabilitation teams in the community and the hospital as described in the text.

Page 12 Data collection section.

Was your topic guide and patient scenarios related to or informed by the programme theory from the realist review. If not, why not?

REPLY: The focus groups were performed alongside the realist review, so the final programme theories had not been developed when the topic guides and patient scenarios were written. The topic guides and framework for analysis were informed by the findings from initial theories, and this has

been clarified in the text.

At present these focus groups sound like a separate project that does not somehow relate to the realist review and survey.

REPLY: The identified themes contributed to the final programme theories as mentioned in the development of the intervention section. The focus groups with patients and their carers highlighted unmet information needs about the process of recovery, the availability of services that patients were entitled to access but were not necessarily aware of, and geographic variation in the provision of services. We have added to the text to clarify how the three work packages linked together to inform intervention development, as well as adding further information on focus group themes and supporting quotes (Table 1).

Page 13: Analysis, credibility and plausibility section.

I am very confused here.

How does thematic analysis using a framework relate to the realist logic of analysis set out by Pawson and Tilley, CMO configurations and programme theory (if at all)?

If they do, an illustrative example of how the data analyses that involves data from the realist review, surveys and focus groups should be provided to aid the reader in understanding the analytic processes used.

This might be focused on a section of the programme theory and could be provided as a supplementary file if preferred - either in the methods or results section.

Such an example would be of great help to those interested in the methodology you have used to develop your intervention.

REPLY: The title of the paper is misleading and has been changed to improve clarity. 'Development of an evidence-based complex intervention for community rehabilitation of hip fracture patients using realist review, survey and focus groups'. This more accurately describes how we used three complementary methods to develop this complex intervention. The themes identified in the focus groups contributed to the final programme theories.

Page 13 Development of the intervention

Again as for the section above on Analysis, plausibility and credibility, it would be helpful if you provide an illustrative example.

REPLY: The emergent themes encompassed a range of experiences and insights that were used to inform the development of the study intervention and further refine the theory areas of the realist review. In particular, they emphasised the patient need for information following fracture that led to the development of an information workbook to be used as part of the study intervention, the content of which was partially based on information requirements that were detailed in focus groups. The variability of care provision was an important factor in both patient and professional perspectives of what constitutes a good package of rehabilitation and the decision to provide six additional therapy sessions to intervention group participants aimed to address this issue.

Page 14 onwards: Results section.

A lot more detail is needed in this section on the results of the various methods used. Of course I realise that you have provided some detail in Figure 1, but a bit more is needed.

For the realist review, I would have expected details on the search processes, screening processes, document characteristics, a PRISMA style flow diagram of disposition of studies etc.

I would strongly suggest you consult the RAMESES publication guidelines for realist syntheses for more details.

The same would go for the surveys and focus groups - more basic details needed for the findings

section.

Of course I realise word count can be an issue, so these may be provided as supplementary files if preferred.

REPLY: This paper aims to summarise the three methodological strands involved in the development of the rehabilitation intervention. A detailed account is in the HTA final report including PRISMA flow diagram, RAMESES checklist etc.

Page 14 Programme theory 1 (Box).

What you have conceptualised as functioning as a mechanism in this box is not a mechanism. It is an intervention strategy (i.e. something that you do).

There is also very limited dis-aggregation of the CMOs.

There is more than one context listed here, one 'mechanism' and at least 4 outcomes.

It's hard to tell what you need to change context-wise to get the outcomes you want and also what the causal mechanism(s) are.

I would suggest you look at the following resources to see if you might be able to re-analyse the data so that a realist logic of analysis has clearly applied:

- RAMESES Training materials

http://www.ramesesproject.org/media/Realist_reviews_training_materials.pdf

- Astbury B, Leeuw F. Unpacking Black Boxes: Mechanisms and Theory Building in Evaluation. American Journal of Evaluation 2010;31:363-81.

- Dalkin S, Greenhalgh J, Jones D, Cunningham B, Lhussier M. What's in a mechanism? Development of a key concept in realist evaluation. Implementation Science 2015;10.

REPLY: Thank you for the references. Our aims were to develop a complex rehabilitation intervention using three complementary methods. The realist review was used to identify the important components of this intervention. We have summarised our findings here and our interpretation of the mechanisms for this purpose. More explanatory detail is in the HTA final report.

Page 14 line 41 to Page 15 line 44:

What would help me in terms of transparency would be if you would please provide for each Programme theory, the data you have used to develop your arguments that underpin the CMOs within your programme theory.

At present the data is quite descriptive and it is hard to see how the data in this section relates to the Programme theory above.

Page 16. Programme theory 2.

The comments I have made for Programme theory 1 above applies to Programme theory 2 as well.

Page 17. Programme theory 3.

The comments I have made for Programme theory 1 above applies to Programme theory 3 as well.

REPLY: This paper has summarised our findings. A detailed account is in the HTA final report.

Page 19 Designing a rehabilitation intervention.

My comments from Page 13 (Development of the intervention) applies to this section.

I found it hard to follow from your interpretations and arguments from data to programme theory. You then also mention a logic model in this section.

How does the logic model relate to the programme theories above and your 'Overarching theory' from Figure 1?

I note for example that patients will be given a "patient-held information workbook". However I was not clear where the data was in your realist review, focus groups and/or surveys to explain why this workbook would 'work' better than anything else for some patients vs other patients or in what contexts.

It is this kind of detail analysis that I would have expected to see in your analysis.

REPLY: Thank you, we have added detail on survey and focus group data and how this was integrated into the intervention development, as well as clarifying how the three work packages fed into one another.

Page 19 Discussion section.

You may wish to reconsider revising the Discussion section in light of my comments on the methods and result sections above as well as the RAMESES publication standards for realist syntheses: <http://bmcmmedicine.biomedcentral.com/articles/10.1186/1741-7015-11-21>

REPLY: This paper has summarised our findings. A detailed account is in the HTA final report.

Page 30 Figure 1

This is an excellent diagram that provides a lot of information very clearly in a highly accessible way. I have a number of observations of this diagram:

a) You might wish to call you "Overarching working theory" your "Initial or rough programme theory". The programme theories below are then sub-components of programme theory.

REPLY: Thank you, we have added the overarching theory to the text to improve clarity. We have not re-named the overarching theory to avoid confusion with our candidate programme theories.

b) What you have assigned as functioning as mechanism in the "Overarching working theory" box are not mechanisms - they are intervention strategies. In other words, things that you would want to do to change contexts so that correct mechanisms are triggered to cause the multiple outcomes you have listed.

REPLY: Our aims were to develop a complex rehabilitation intervention. The realist review was used to identify the important components of this intervention. We have summarised our findings here and our interpretation of the mechanisms for this purpose.

I hope my comments are of help and good luck with your revisions.

Reviewer: 6

Reviewer Name: Joanne Greenhalgh

Institution and Country: School of Sociology and Social Policy

Please state any competing interests: None

Please leave your comments for the authors below

I really like what this paper is trying to do and I think it is entirely appropriate to utilize realist methods to develop a complex intervention. I enjoyed reading the paper and it really made me think. It reports on an impressive amount of work and I support the aspirations of the authors. There is certainly a need for someone to explicate the methodology through which realist methods can be used to underpin the development of a complex intervention.

REPLY: Thank you for your positive feedback.

The paper is ambitious (perhaps too ambitious) in its scope. I wasn't quite certain whether the paper was focused on presenting findings or explaining/explicating a methodology. The title suggests the latter - it suggests that the paper is going to explain how realist methods were used to develop a

complex intervention. However, my overall feeling at the end of the paper is that there was more focus on providing information about the individual trees and much less on explaining how they work together to form a wood. In other words, much of the methods section of the paper focuses on explaining how each separate data set was assembled and analysed but there is much less discussion of how and through what processes all these data were integrated through realist logic to underpin intervention development. The results section does attempt this integration, but as a reader I felt that the potential richness of each data set disappeared and the realist logic underpinning the integration of these data got a bit lost. I would have liked to have seen more detail regarding the initial programme theories and how each data set has contributed to testing and refining these. It's not quite clear whether the theories represent 'refined theories' - ie an 'end product' or 'theories that then require further testing using realist logic', ie for example, when intervention is further evaluated.

REPLY: The title was misleading and has been clarified. The realist review was one component of intervention development along with the survey and the focus groups. More detail concerning each programme theory is available in the final report. These theories were used as a means to define the components of the rehabilitation intervention.

I appreciate the challenge the authors (I imagine) have faced in writing up their work because it is difficult to both provide sufficient detail about the individual pieces of data collection at the same time as provide a more 'meta-view' of how these individual pieces have been integrated. I wonder therefore whether there is merit in the idea of either allowing the authors more words to do this, or perhaps putting some of the detail regarding data collection and analysis in a box/supplementary file along with the manuscript. The more radical alternative is to suggest writing up a realist review in a separate paper and then in a further paper explain how its findings were tested and refined via focus groups and placed in the context of the findings of the survey to develop an intervention. However, I do think with some revision and restructuring it is possible to achieve this within a single paper.

REPLY: Some more detail has been added, but this paper has summarised our findings. A detailed account is in the HTA final report.

I have made some comments on specific sections of the paper in the hope the authors might find them helpful in revising the paper:

1. In the methods section, it would be helpful to have a section which explains how each strand of data collection related to the other and how they each informed the process of theory testing and refinement. It would be useful to know what the role of each strand of data collection was - it seems to me that the survey was not really used for theory testing but to provide information on current context - so how was this integrated with the review and focus groups? The diagram (Figure 1) does not really show the dynamics of this process. For example, it's not quite clear whether the three different strands were conducted independently and the integration process occurred only at the end (if so - how was this done?) or whether integration occurred throughout the process, around the process of theory gleaning, testing and refining, in keeping with realist logic.

REPLY: Thank you. We have provided more detail in the methods to improve clarity. The purpose of the survey was to provide context.

2. When each item of data collection is described, I would have liked to have seen a sentence of two which explains how data collection and analysis was used to glean, test or consolidate theories. This is especially the case for the focus groups. To use Manzano's (2016) typology - were the focus groups about gleaning, testing or refining theory - or all three? How did the focus groups relate to the realist review - were they conducted after or alongside the review and how did the findings of the review inform the conduct of the focus groups? The section on in the methods on 'development of the

intervention' needs expanding to give more detail in the integration process and the role of theory in this process.

REPLY: More detail has been added to improve clarity. A scoping search of systematic reviews, guidelines and theoretically rich primary studies was performed to map out the important areas and research gaps. This generated a list of questions, which could be grouped under different domains relating to: patients, health care and rehabilitation teams, rehabilitation programmes and the settings in which rehabilitation was delivered. These were used to develop the questions for the survey and topic guides for the focus groups. The initial framework for the focus group analysis was broadly developed from the theory areas identified as important to guide the realist review and it was used to index the transcripts.

3. What is the status of the three programme theories? As discussed above, are they an 'end product' or do they represent a 'mid point' in the theory testing process - ie hypotheses that can then be tested when the intervention is further evaluated?

REPLY: These theories were used as a means to define the components of the rehabilitation intervention. The acceptability and feasibility of this intervention was then tested in a phase II feasibility study, which is not reported in this paper. The effectiveness of the intervention will be tested in a future RCT.

4. I don't want to come over as the 'CMO police' but my overall sense when reading the PT, as presented in the boxes, is that they focus much more on delineating the 'resources' element of mechanisms and much less on the 'responses' aspect. There is more detail in the narrative text surrounding each, but I did want to ask (for example, in relation to PT 1)- so what is it about tailoring that enables these desired outcomes to occur? Maybe it is just about signposting this more clearly - for example, the sentence that really seemed to come close to grasping the 'response' element was the findings from the focus groups that indicated patients found it hard to engage with generalised goals and preferred individualised ones - this begs the intriguing question of - why? Furthermore, the explication of context seemed to represent the 'existing context' - ie the problem that the intervention is trying to solve, rather than hypothesising how possible variations in context might support or constrain the different mechanisms through which rehabilitation is expected to work. This is fine and its often how PT theories start, but it does suggest the further testing will be needed to/result in a more detailed understanding how variations in context might shape the way the intervention works. Again, would be worth signposting this.

REPLY: This is a summary of the realist review. More detail is available in the final report. The response element of tailoring is that it involves collaborative decision making and goal-setting as described in the text. The context of the tailoring programme theory is the degree of fit between the variation in need and the rehabilitation programme provided.

5. In the final section of the results, Figure 2 does a good job of connecting the programme theories and the intervention itself. It would be helpful if these connections were made more clearly in the text narrative itself.

REPLY: Thank you. We have added some more detail throughout on results from the survey and focus groups, how these were integrated with the realist review to develop the intervention and how the three work packages fed into each other.

VERSION 2 – REVIEW

REVIEWER	Katherine Harding Eastern Health, Victoria, Australia
REVIEW RETURNED	13-Jul-2017

GENERAL COMMENTS	I am satisfied that the concerns raised in my original review have been adequately addressed by the authors. The paper is now much clearer to read, and the strands of the different methods of data collection now come together in much more clear and meaningful way. The paper is still quite lengthy, but I feel now that the depth of the content now justifies the length if this can be accommodated from an editorial perspective.
---

REVIEWER	Paula M van Wyk University of Windsor Canada
REVIEW RETURNED	10-Jul-2017

GENERAL COMMENTS	I really appreciate the authors substantial edits to the manuscript and their careful attempt to address the majority of issues posed by the numerous reviewers of the paper. I understand that the authors attempted to make the paper focus on the development of a complex intervention and that the realist review was used to contribute to the development of the intervention. Unfortunately, I fear that the focus of the paper throughout still seems to change between whether it is novel use of the realist review or the development of a programme. It is my suggestion that the focus is on the realist review approach as that is the novel aspect of this paper and the state future directions even discuss how the realist review can be used (as opposed to the programme). What I believe is further support for the focus on the realist approach is that the findings are mostly repetitive of information already published. I do believe that this paper has merit and can further the literature – but I believe the most effective way to do so is by focusing on the realist approach that was used with the development of the programme as the context for which you employed the approach. - Findings state that the review indicated a detailed assessment of patient's pre-fracture level of function, current cognitive status and other co-morbid conditions. Furthermore should involve decision making by the patients and caregivers. All with active engagement to help tailor the programme. Agreed. As does much of the literature. What is needed is the best way to actually conduct these assessments and incorporate the involvement of the patient and caregivers. - New finding – that less than 50% of OTs included anxiety
--

	managements and developing self-awareness – this is important to report/highlight. - I think the information regarding the need to include psychological components in rehabilitation interventions is something to highlight. Although this findings provides support for information already published [i.e. 92] it is one of less highlighted or repetitive findings. Your other findings, no offense intended, are not novel – they have been in the literature numerous times. - I am hesitant to agree that you have achieved the goal of building an explanatory account of the mechanisms behind rehab programmes through the realist review approach – but I did like the approach. You did not establish which components were considered to be effective and in which circumstances – so I would delete that aspect of your statement. (page 28) There is still a lot of value laden language in the paper, I strongly advise that this is corrected. There are also a number of grammatical issues. I accept the reason as to why the review of the literature component only spanned up until 2013 – I would appreciate a caveat in the manuscript, something like: Twenty one databases were searched from inception until Feb 2013 in order to be used as the next step of programme development (Appendix 1). Thank you for adding the appendix by the way. I am disappointed that the details explaining “rich” and “thick” were not added, but I understand there are restrictions of word limit. Please note – Figure 2 as presented on Page 2 was not legible. I very much appreciated the insertion of Table 1. This table provides some rich data, that I did wish was exploded in further depth if the focus of the paper is to be on why certain components of the programme were viewed as vital.
--	--

REVIEWER	Dr Margaret Coulter Smith Queen Margaret University Scotland
REVIEW RETURNED	06-Jul-2017

GENERAL COMMENTS	This paper illustrates the application of a realist review, survey and focus groups to the development of a complex intervention for rehabilitation post hip fracture. The work is located within Phase 1 of the MRC Complex Interventions Framework and represents an innovative way to develop a multi-component complex intervention that would clearly require further testing and refining in subsequent work. The authors provide good evidence that the work is well grounded in the theoretical and research literature.
--

REVIEWER	Geoff Wong University of Oxford, United Kingdom
REVIEW RETURNED	30-Jun-2017

GENERAL COMMENTS	I found the explanations provided in the highly detailed response to reviewers comments very helpful. I did however note that much of the text in red in this document never made it to the revised manuscript. I appreciate that there was both an ask for more detail and also less, my reviewer comments having fallen into the former camp. This begs the question of what is a reasonable amount of detail to present to the reader so that they can make up their own minds about the rigour of three work streams. Personally, for the realist review, I felt that more (and not less) detail was needed. For me, this would include more details about the actual analyses - from raw data through to interpretations and inferences made. As such, many of my initial comments remain unaddressed. I do realise that the authors make the argument that there is a more detailed report for the project for the funder and so it may be that signposting readers who are interested in the nitty-gritty of each work stream may be a reasonable compromise. However, if this is the case then a required major revision to this manuscript is that readers are signposted to the location of this detailed report AND that the reader is told that what is presented in this manuscript are overview summaries of the findings. Minor points that I felt needed addressing are:  - Page 14 11 to 20: Whilst this is a minor point, I would strongly encourage you to remove the (C), (M)s and (O) in this paragraph. The (M)s labelled in this paragraph are not mechanisms, but intervention strategies. It's still perfectly OK to initially start off with a working theory that is not realist in nature. (The above also applies to the text in Figure 1). - Page 14: lines 31 to 46: I would strongly encourage you to remove the (C), (M) and (O) in this box about Programme theory 1. The (M) labelled in this paragraph is not a mechanism. - Page 18 lines 8 to 15: I would strongly encourage you to remove the (C), (M) and (O) in this box about Programme theory 2. The (M) labelled in this paragraph is not a mechanism. - Page 20 lines 15 to 23 I would strongly encourage you to remove the (C), (M) and (O) in this box about Programme theory 2. The (M) labelled in this paragraph is not a mechanism.
---

REVIEWER	Joanne Greenhalgh University of Leeds, UK
REVIEW RETURNED	10-Jul-2017

GENERAL COMMENTS	The authors have gone to great lengths to address the comments
--

	from a large number of reviewers. In my view, this is an important paper because it illustrates the process of developing a complex intervention using realist methods. There is no 'perfect' or 'ideal' way to do this and the paper is likely to attract a good deal of discussion once out in the public domain.
--	---

VERSION 2 – AUTHOR RESPONSE

Reviewer: 1

Reviewer Name: Katherine Harding

Institution and Country: Eastern Health, Victoria, Australia

Please state any competing interests: None declared

Please leave your comments for the authors below

I am satisfied that the concerns raised in my original review have been adequately addressed by the authors. The paper is now much clearer to read, and the strands of the different methods of data collection now come together in much more clear and meaningful way. The paper is still quite lengthy, but I feel now that the depth of the content now justifies the length if this can be accommodated from an editorial perspective.

REPLY: Thank you

Reviewer: 2

Reviewer Name: Paula M van Wyk

Institution and Country: University of Windsor, Canada

Please state any competing interests: None declared

Please leave your comments for the authors below

I really appreciate the authors substantial edits to the manuscript and their careful attempt to address the majority of issues posed by the numerous reviewers of the paper. I understand that the authors attempted to make the paper focus on the development of a complex intervention and that the realist review was used to contribute to the development of the intervention.

REPLY: Thank you

Unfortunately, I fear that the focus of the paper throughout still seems to change between whether it is novel use of the realist review or the development of a programme. It is my suggestion that the focus is on the realist review approach as that is the novel aspect of this paper and the state future directions even discuss how the realist review can be used (as opposed to the programme). What I believe is further support for the focus on the realist approach is that the findings are mostly repetitive of information already published. I do believe that this paper has merit and can further the literature – but I believe the most effective way to do so is by focusing on the realist approach that was used with the development of the programme as the context for which you employed the approach.

- Findings state that the review indicated a detailed assessment of patient's pre-fracture level of function, current cognitive status and other co-morbid conditions. Furthermore should involve decision making by the patients and caregivers. All with active engagement to help tailor the programme. Agreed. As does much of the literature. What is needed is the best way to actually conduct these assessments and incorporate the involvement of the patient and caregivers.

- New finding – that less than 50% of OTs included anxiety managements and developing self-awareness – this is important to report/highlight.

REPLY: Thank you. This has been highlighted in the discussion.

- I think the information regarding the need to include psychological components in rehabilitation interventions is something to highlight. Although this findings provides support for information already published [i.e. 92] it is one of less highlighted or repetitive findings. Your other findings, no offense intended, are not novel – they have been in the literature numerous times.

REPLY: Thank you. This has been highlighted in the discussion.

- I am hesitant to agree that you have achieved the goal of building an explanatory account of the mechanisms behind rehab programmes through the realist review approach – but I did like the approach. You did not establish which components were considered to be effective and in which circumstances – so I would delete that aspect of your statement. (page 28)

REPLY: This has been deleted.

There is still a lot of value laden language in the paper, I strongly advise that this is corrected. There are also a number of grammatical issues.

REPLY: We have amended the abstract accordingly.

I accept the reason as to why the review of the literature component only spanned up until 2013 – I would appreciate a caveat in the manuscript, something like: Twenty one databases were searched from inception until Feb 2013 in order to be used as the next step of programme development (Appendix 1). Thank you for adding the appendix by the way.

REPLY: This caveat has been added.

I am disappointed that the details explaining “rich” and “thick” were not added, but I understand there are restrictions of word limit.

Please note – Figure 2 as presented on Page 2 was not legible.

I very much appreciated the insertion of Table 1. This table provides some rich data, that I did wish was exploded in further depth if the focus of the paper is to be on why certain components of the programme were viewed as vital.

Reviewer: 4

Reviewer Name: Dr Margaret Coulter Smith

Institution and Country: Queen Margaret University, Scotland

Please state any competing interests: None declared.

Please leave your comments for the authors below

This paper illustrates the application of a realist review, survey and focus groups to the development of a complex intervention for rehabilitation post hip fracture. The work is located within Phase 1 of the MRC Complex Interventions Framework and represents an innovative way to develop a multi-

component complex intervention that would clearly require further testing and refining in subsequent work. The authors provide good evidence that the work is well grounded in the theoretical and research literature.

REPLY: Thank you

Reviewer: 5

Reviewer Name: Geoff Wong

Institution and Country: University of Oxford, United Kingdom

Please state any competing interests: None declared

Please leave your comments for the authors below

Thank you for asking me to re-review this manuscript.

I found the explanations provided in the highly detailed response to reviewers comments very helpful. I did however note that much of the text in red in this document never made it to the revised manuscript.

I appreciate that there was both an ask for more detail and also less, my reviewer comments having fallen into the former camp. This begs the question of what is a reasonable amount of detail to present to the reader so that they can make up their own minds about the rigour of three work streams.

Personally, for the realist review, I felt that more (and not less) detail was needed. For me, this would include more details about the actual analyses - from raw data through to interpretations and inferences made. As such, many of my initial comments remain unaddressed.

I do realise that the authors make the argument that there is a more detailed report for the project for the funder and so it may be that signposting readers who are interested in the nitty-gritty of each work stream may be a reasonable compromise. However, if this is the case then a required major revision to this manuscript is that readers are signposted to the location of this detailed report AND that the reader is told that what is presented in this manuscript are overview summaries of the findings.

REPLY: Statements to this effect have been added to the beginning of the methods and the results.

Minor points that I felt needed addressing are:

- Page 14 11 to 20:

Whilst this is a minor point, I would strongly encourage you to remove the (C), (M)s and (O) in this paragraph. The (M)s labelled in this paragraph are not mechanisms, but intervention strategies. It's still perfectly OK to initially start off with a working theory that is not realist in nature.

(The above also applies to the text in Figure 1).

- Page 14: lines 31 to 46:

I would strongly encourage you to remove the (C), (M) and (O) in this box about Programme theory 1. The (M) labelled in this paragraph is not a mechanism.

- Page 18 lines 8 to 15:

I would strongly encourage you to remove the (C), (M) and (O) in this box about Programme theory 2. The (M) labelled in this paragraph is not a mechanism.

- Page 20 lines 15 to 23

I would strongly encourage you to remove the (C), (M) and (O) in this box about Programme theory 2. The (M) labelled in this paragraph is not a mechanism.

REPLY: These letters have been removed.

Good luck with the rest of the project.

Reviewer: 6

Reviewer Name: Joanne Greenhalgh

Institution and Country: University of Leeds, UK

Please state any competing interests: None declared

Please leave your comments for the authors below

The authors have gone to great lengths to address the comments from a large number of reviewers. In my view, this is an important paper because it illustrates the process of developing a complex intervention using realist methods. There is no 'perfect' or 'ideal' way to do this and the paper is likely to attract a good deal of discussion once out in the public domain.

VERSION 3 – REVIEW

REVIEWER	P.M. van Wyk University of Windsor Canada
REVIEW RETURNED	22-Aug-2017

GENERAL COMMENTS	Thank you for taking the time to address reviewers concerns. Although I feel some concerns should still be address - I do not want to hold this paper up from being published.
--

REVIEWER	Geoff Wong University of Oxford, United Kingdom
REVIEW RETURNED	22-Aug-2017

GENERAL COMMENTS	Thank you for considering my comments and making the necessary changes. One last revision is needed to Figure 1 - the labels (C), (M) and (O) need to be removed as has been done in the main text of the manuscript.
--